# Nitric oxide radicals are emitted by wasp eggs to kill mold fungi

Erhard Strohm[1]*, Gudrun Herzner[1], Joachim Ruther[2], Martin Kaltenpoth[1,3†], Tobias Engl[1,3†]

[1]Evolutionary Ecology Group, Institute of Zoology, University of Regensburg, Regensburg, Germany; [2]Chemical Ecology Group, Institute of Zoology, University of Regensburg, Regensburg, Germany; [3]Insect Symbiosis Research Group, Max Planck Institute for Chemical Ecology, Jena, Germany

**Abstract** Detrimental microbes caused the evolution of a great diversity of antimicrobial defenses in plants and animals. Insects developing underground seem particularly threatened. Here we show that the eggs of a solitary digger wasp, the European beewolf *Philanthus triangulum,* emit large amounts of gaseous nitric oxide (NO·) to protect themselves and their provisions, paralyzed honeybees, against mold fungi. We provide evidence that a NO-synthase (NOS) is involved in the generation of the extraordinary concentrations of nitrogen radicals in brood cells (~1500 ppm NO· and its oxidation product $NO_2$·). Sequencing of the beewolf *NOS* gene revealed no conspicuous differences to related species. However, due to alternative splicing, the NOS-mRNA in beewolf eggs lacks an exon near the regulatory domain. This preventive external application of high doses of NO· by wasp eggs represents an evolutionary key innovation that adds a remarkable novel facet to the array of functions of the important biological effector NO·.
DOI: https://doi.org/10.7554/eLife.43718.001

**\*For correspondence:**
erhard.strohm@ur.de

**Present address:** †Evolutionary Ecology, Institute of Organismic and Molecular Evolution, Johannes Gutenberg University, Mainz, Germany

**Competing interests:** The authors declare that no competing interests exist.

## Introduction

Microbes pose a major threat to the health of all animals and plants. These have responded by evolving a great diversity of defenses including hygienic behaviors (*Gilliam et al., 1983*), antimicrobial chemicals (*Herzner et al., 2013*; *Vilcinskas et al., 2013*; *Gross et al., 2008*), complex immune systems (*Hooper et al., 2012*; *Iwasaki and Medzhitov, 2010*), and defensive symbioses (*Kaltenpoth et al., 2005*; *Flórez et al., 2017*). Besides such pathogenic effects, many bacteria and fungi are severe, but often neglected, competitors of animals for nutrients, thus prompting the evolution of mechanisms to preserve food sources (*Janzen, 1977*; *Rozen et al., 2008*).

Some animals are particularly prone to suffer from microbial attack due to (1) high abundance of potentially harmful microbes in their environment, (2) a microbe-friendly microclimate and/or (3) limited defense mechanisms. The progeny of many insect species develop under warm and humid conditions in the soil, where they are exposed to a high diversity of bacteria and fungi. Moreover, compared to adult insects, immature stages, in particular eggs, have usually reduced abilities to prevent microbial infestation due to, for example, a thin cuticle or an inability to groom (*Wilson and Cotter, 2013*; *Tranter et al., 2014*). The situation is even aggravated when eggs and larvae have to develop on limited amounts of provisions that are susceptible to attack by ubiquitous and fast growing putrefactive bacteria and mold fungi (*Janzen, 1977*; *Arce et al., 2013*).

Such hostile conditions prevail in nests of subsocial Hymenoptera like the European beewolf *Philanthus triangulum* (Hymenoptera, Crabronidae). The offspring of these solitary digger wasps develop in subterranean brood cells provisioned by the female wasps with paralyzed honeybee workers (*Apis mellifera*, Apidae, Hymenoptera) (*Strohm and Linsenmair, 2000*) (*Figure 1A*). The beewolf egg is laid on one of the bees, the larva hatches after three days, feeds on the bees for six

**eLife digest** Humans use heat, cooling, and freezing to protect their foods from mold and bacteria. Many animals, including a wasp called the European beewolf, have also developed ways to store and preserve food. Female beewolves hunt honeybees. After paralyzing a bee, the beewolf takes the body into an underground chamber and lays an egg on it. When a larva hatches from the egg, it feeds on the bee. The warm, humid conditions in the chamber provide ideal conditions in which larvae can develop, but also encourage mold and bacteria to grow.

Previous research has uncovered two methods used by beewolves to fight off mold. In 2007, researchers discovered that the female beewolf coats her bee prey with a layer of fats. This prevents water loss and keeps the outside of the bee dry so that mold spores cannot grow. In 2010, a further study showed that the female beewolf grows helpful bacteria inside her antennae and transfers some to her young. The bacteria produce antibiotics that protect the larvae and their cocoons from mold. But these two strategies alone cannot explain the high survival rate of beewolf young. This suggests that the beewolves have at least one more strategy to prevent mold from growing.

Now, Strohm et al. – including some of the researchers involved in the 2007 and 2010 studies – show that beewolf eggs emit high levels of a gas called nitric oxide, which reacts with oxygen to form nitrogen dioxide. Nitrogen dioxide is part of the air pollution generated by cars and is harmful to many species in high concentrations. Nitric oxide also plays an important role for many biochemical processes in virtually all organisms, albeit in very low concentrations. The beewolf eggs produce comparatively huge amounts of this gas to fumigate their brood chambers and protect themselves and their food from mold.

Strom et al. then investigated how the eggs produce nitric oxide. The eggs appear to use the same enzyme that some other organisms use to produce nitric oxide. However, the wasp version of the enzyme contains a small modification that might explain why the eggs can produce the gas in such large amounts.

Learning more about how beewolves evolved different anti-mold strategies could help researchers to develop new antimicrobial treatments for medical applications. In addition, it is not yet clear how the wasp eggs survive in high concentrations of nitric oxide and nitric dioxide. Inflammation and some human diseases produce nitric oxide, killing nearby cells. Understanding how the beewolf eggs survive could therefore help to treat these cells or protect them from damage.

DOI: https://doi.org/10.7554/eLife.43718.002

to eight days, then spins a cocoon and either emerges the same summer or hibernates. The warm and humid microclimate in the brood cell promotes larval development but also favors fast growing, highly detrimental fungi (*Engl et al., 2016*). Without any countermeasures the provisions will be completely overgrown by mold fungi within three days (*Figure 1B*), and the beewolf larva becomes infested by fungi or starves to death (*Strohm and Linsenmair, 2001*; *Herzner et al., 2011a*).

We have previously documented two adaptations that beewolves have evolved to counter the detrimental effects of fungi on their brood. First, beewolf females reduce molding of the larval provisions by coating the paralyzed bees with ample amounts of unsaturated hydrocarbons (*Herzner et al., 2007*). This embalming changes the physicochemical properties of the preys' surface causing reduced water condensation on the bees (*Herzner and Strohm, 2007*). Due to the deprivation of water, germination and growth of fungi is delayed by two to three days (*Herzner et al., 2011b*). Second, during the long period of overwintering in their cocoons beewolf larvae are protected by antibiotics on their cocoons (*Kaltenpoth et al., 2005*; *Kroiss et al., 2010*). Prior to oviposition, beewolf females apply a secretion containing symbiotic *Streptomyces* bacteria to the ceiling of the brood cell. The secretion is taken up by the larvae and incorporated into the silk threads of their cocoons. There, the bacteria produce several antibiotics that effectively protect the cocoon and, thus, the larvae against fungus infestation (*Kroiss et al., 2010*; *Engl et al., 2018*).

Despite the considerable effect of prey embalming, when removed from brood cells at least 50% of embalmed bees showed fungus infestation within six days after oviposition (*Strohm and Linsenmair, 2001*). Since in natural brood cells only around 5% of the progeny succumb to mold fungi

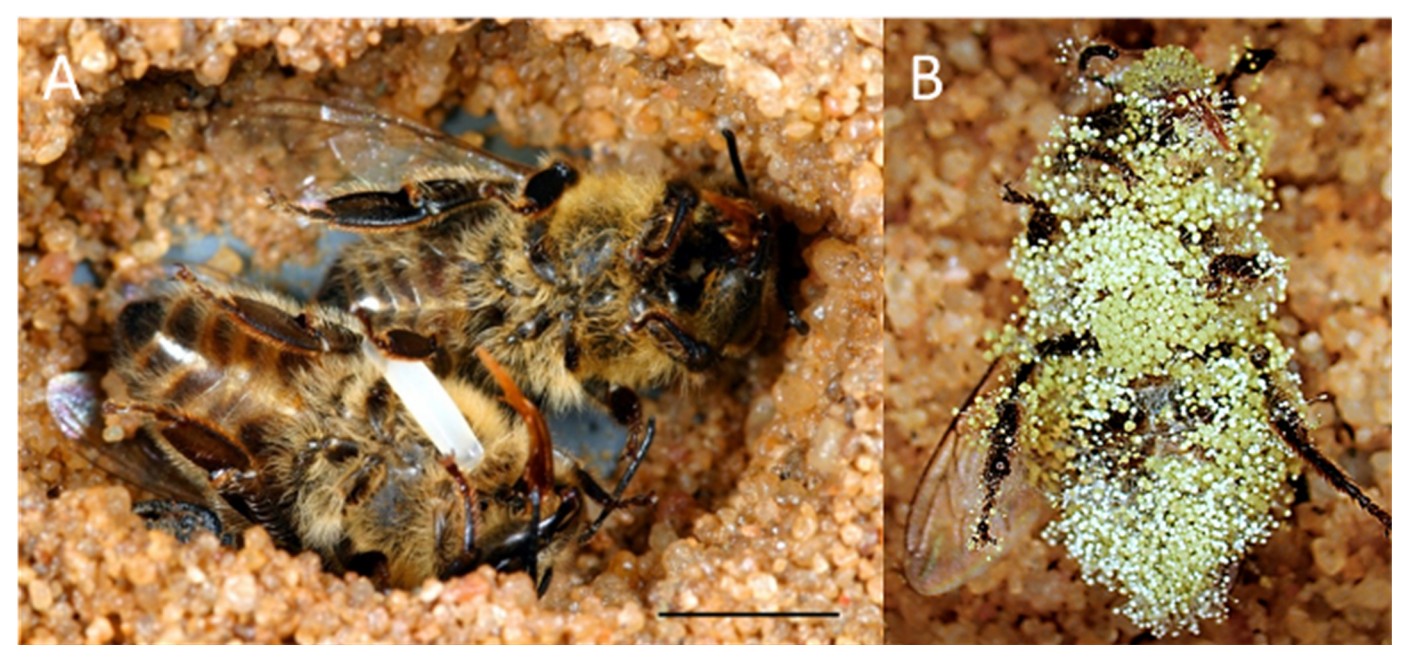

**Figure 1.** Paralyzed honeybees under different conditions. (**A**) Brood cell of the European beewolf with two bees, one carrying an egg, in an observation cage. (**B**) Honeybee paralyzed by a beewolf female but immediately removed and kept in an artificial brood cell, heavily overgrown by mold fungi that have already developed conidia. Scale bar = 5 mm.

DOI: https://doi.org/10.7554/eLife.43718.003

(**Strohm and Linsenmair, 2001**), we searched for an additional antimicrobial defense mechanism that takes effect during the early stages of beewolf development.

Here we report on a unique antifungal strategy that is employed by beewolf eggs to defend themselves and their provisions against mold fungi. Employing bioassays we discovered that beewolf eggs emit a strong antifungal agent that we identified as the gaseous radical nitric oxide (NO·). We characterize the amount, time course and temperature dependence of emission and show that synthetic NO· exerts a similar effect as the gas emitted by beewolf eggs. Furthermore, we tested whether there was an interaction of the gas emitted by the eggs and the embalming of the prey by beewolf females. Using histological methods, inhibition assays, and gene expression analysis, we elucidate a biosynthetic pathway for NO· synthesis in beewolf eggs. To explore the evolutionary background of this remarkable antimicrobial strategy, we sequenced the relevant gene and mRNA. Our findings reveal a novel function of the eminent and widespread biological effector NO· in providing an extended immune defense to the producer by sanitizing its developmental microenvironment.

## Results

### Emission of an antifungal volatile by beewolf eggs

Thorough examination of beewolf nests in observation cages (**Strohm and Linsenmair, 1994**) revealed that within 24 hr after oviposition, a conspicuous pungent smell occurred that was clearly emanating from the eggs and disappeared by the time the larvae hatched. We hypothesized that this smell was due to an antifungal agent. When paralyzed honeybees from completed beewolf brood cells were incubated individually, bees carrying an egg showed significantly delayed fungus growth compared to bees without egg over the period from oviposition to cocoon spinning (Kaplan Meier survival analysis, Breslow test, day 0–11: Chi square = 12, df = 1, p=0.001; *Figure 2A*). This difference was also significant for the period from oviposition to the hatching of the larvae (day 0–3: Chi square = 9.5, df = 1, p=0.002), suggesting that this effect is not due to possible antifungal mechanisms of the larvae but that it is mediated by the egg. Considering the distinctive odor that

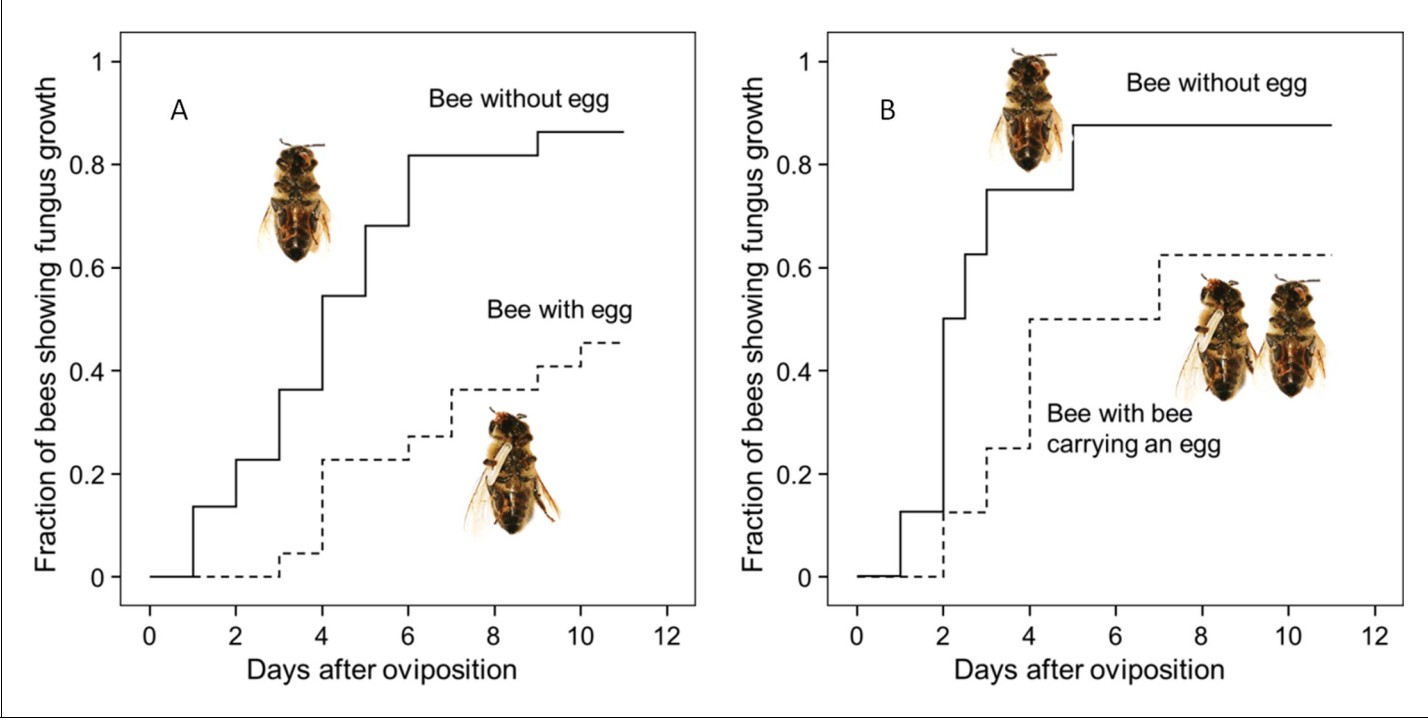

**Figure 2.** Onset of fungal growth on paralyzed honeybees taken from *Philanthus triangulum* nests and kept in artificial brood cells. The fraction of bees showing first signs of fungal growth is shown as a function of days since oviposition. (**A**) Honeybees that either carried an egg (dashed line) or not (solid line) (N = 22 each, hazard ratio = 0.29, 95% confidence interval: 0.13–0.64). (**B**) Honeybees that were either kept alone (solid line) or shared a brood cell with a bee carrying an egg (dashed line) (N = 16 each, hazard ratio = 0.39, 95% confidence interval: 0.17–0.9).

DOI: https://doi.org/10.7554/eLife.43718.004

The following source data is available for figure 2:

**Source data 1.** Effect of egg on fungus growth.

DOI: https://doi.org/10.7554/eLife.43718.005

emanated from the eggs, we tested whether the antifungal effect is caused by a volatile agent. Two experiments supported this assumption. First, provisioned bees without wasp eggs that were kept in artificial brood cells together with bees carrying an egg (but without physical contact) showed significantly delayed fungal growth compared to control bees that were kept alone (Breslow test, day 0–11: Chi square = 7.6 df=1, p=0.006; day 0–3: Chi square = 9.1, df = 1, p=0.003; *Figure 2B*). Second, when one of the most abundant mold species from infested beewolf brood cells, the fast growing *Aspergillus flavus* (*Engl et al., 2016*), was exposed to the volatiles presumably emanating from beewolf eggs on nutrient agar for three days, its growth was entirely inhibited, whereas it thrived in controls (for all observation times 24 hr, 48 hr, 72 hr: binomial test: N = 20, p<0.001, *Figure 3*). Notably, when the beewolf larvae were removed from the assays shortly after hatching (three days after oviposition), no fungal growth occurred in the exposed areas during another three days. A similar experiment showed that paralyzed honeybees alone did not show any antifungal activity (Appendix 1: Additional data 1). Analogous bioassays with five other fungal strains (*A. flavus* strain B, *Mucor circinelloides*, *Penicillium roqueforti*, *Candida albicans* and *Trichophyton rubrum*) revealed that in all cases fungus growth was likewise completely inhibited in the area that was exposed to volatiles from beewolf eggs, whereas the fungi thrived in the respective control areas (for each strain: N = 8, binomial test: p<0.01). We conclude that beewolf eggs release a volatile compound with broad spectrum fungicidal properties.

## Identification of the antifungal volatile

The odor emanating from the eggs was similar to that of strong oxidants like chlorine, ozone and nitrogen dioxide (subjective evaluation of several observers, [*Mücke and Lemmen, 2010*]). In fact,

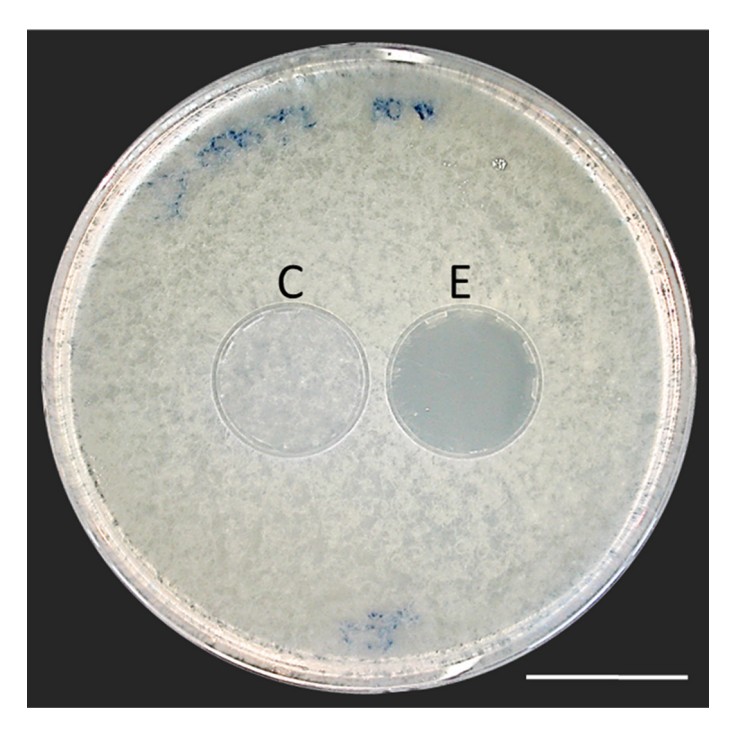

**Figure 3.** Bioassay demonstrating the inhibitory effect of a beewolf egg against *Aspergillus flavus*. Two areas on the agar were covered by caps of a volume similar to natural beewolf brood cells. One cap, the control (C), was empty, while the experimental cap (E) contained a fresh beewolf egg attached to the ceiling of the cap. The caps were removed and the picture was taken after 24 hr of incubation at 25°C. The control area (C) shows dense whitish fungal hyphae similar to the surroundings. However, the area that was exposed to the volatiles from a beewolf egg (E) shows bare agar, indicating that the growth of this aggressive fungus was entirely inhibited. Scale bar = 2.5 cm.
DOI: https://doi.org/10.7554/eLife.43718.006

the generation of a strong blue coloration when placing a beewolf egg into the lid of a reaction tube filled with a iodide/starch solution ([*Jander and Blasius, 1971*], see iodometry in Materials and methods) revealed the existence of an oxidant in the headspace of beewolf eggs. There are few gaseous oxidants that might be considered, in particular chlorine, ozone and nitrogen dioxide. Ozone has, to our knowledge, not been described to be synthesized by organisms. Molecular chlorine has been reported as an intermediate in some organisms but its occurrence seems to be restricted to phagocytosis (*Hazen et al., 1996*). The most likely candidate was the radical nitrogen dioxide ($NO_2$), because there is a plausible way for it being generated by wasp eggs: Insect embryos synthesize small amounts of nitric oxide ($NO$) as signaling effectors for developmental processes (*Andersen et al., 2013*). If such odorless $NO$ was emitted from the egg, it would spontaneously react with oxygen (*Soegiarto et al., 2003*; *Mur et al., 2011*) to yield the strong-smelling $NO_2$. Moreover, belonging to the reactive nitrogen species (RNS), $NO$ and $NO_2$ show considerable antimycotic activity (*Fang, 1997*; *Lai et al., 2011*) that would explain the observed fungicidal effect of beewolf eggs. Hence, we hypothesized that eggs synthesize and emit $NO$ that reacts with the oxygen in brood cells to $NO_2$ thus generating the pungent smell and the antimycotic activity.

We tested whether beewolf eggs produce and emit $NO$ and/or $NO_2$ by conducting a series of experiments. First, headspace samples of confined beewolf eggs were subjected to the Griess assay, the standard procedure for the specific detection of $NO$ and $NO_2$ (*Tsikas, 2007*). The emerging red color of the resulting azo dye (*Figure 4—figure supplement 1*) clearly indicated the presence of $NO$/$NO_2$. To visualize the emission of $NO$ from beewolf eggs, we sprayed a solution of an $NO$ specific fluorescent probe, Diaminorhodamin-4M AM (DAR4M-AM), onto prey bees carrying freshly laid eggs. The small droplets of the DAR4M-AM solution on the bees showed a clear fluorescence

around the egg that increased over several hours (*Figure 4B*). No such effect was seen on control bees without eggs (*Figure 4A*). Moreover, beewolf eggs injected with the DAR4M-AM solution showed a strong fluorescence that peaked about one day after oviposition (N = 45, *Figure 5A*). The same treatment yielded only weak fluorescence in the eggs of two other Hymenoptera (the Emerald cockroach wasp, *Ampulex compressa*, N = 9, and the Red mason bee, *Osmia bicornis*, N = 12; *Figure 5C and D*) and in newly hatched beewolf larvae (N = 4, *Figure 5—figure supplement 1*). Autofluorescence of beewolf eggs injected with buffer only (N = 10) was negligible (*Figure 5B*). These findings strongly imply that beewolf eggs produce and release NO˙.

## Amount and time course of NO˙ emission

Using iodometry, we determined that a beewolf egg (volume: $4.1 \pm 0.5$ mm$^3$, N = 16) emits on average $0.25 \pm 0.09$ µmol NO˙ (N = 233). The rate of NO˙ production was initially very low, but increased to a distinct peak 14–15 hr (at 28°C) after oviposition (*Figure 6*); around 90% of NO˙ emission occurred within a two-hour period. Assuming no loss due to reactions or leaking out of the confined space of brood cells (volume $3.2 \pm 0.9$ cm$^3$, N = 250), the nitrogen oxides would accumulate to average concentrations of $1690 \pm 680$ ppm. The timing of the onset of NO˙ emission was strongly temperature dependent (*Figure 6—figure supplement 1*), with higher temperatures resulting in an earlier NO˙ production (temperature coefficient $Q_{10}$ = 2.74).

## Antifungal effect of synthetic NO˙

To test whether the observed antifungal effect of the gas that is emitted by beewolf eggs was due to NO˙, we conducted an experiment using synthetic NO˙ but otherwise emulated natural conditions as closely as possible. Artificial brood cells containing a bee without egg were injected with either synthetic NO˙ to generate a peak concentration of 1500ppm or with nitrogen as controls. There was a significant delay in the onset of fungus growth on the bees exposed to synthetic NO˙ as compared to controls (*Figure 7*, Breslow test: Chi square = 13.3, df = 1, p<0.0001).

## Combined effect of NO˙ emission and prey embalming

Since both, NO˙ emission and embalming of the prey bees by beewolf females take effect during the first days after oviposition, we assessed the antifungal effects of these defense mechanisms alone and in combination. Bioassays with bees in artificial brood cells that were either embalmed or not and carried an egg or not revealed that the onset of fungal growth differed significantly among treatment groups (*Figure 8*: Kaplan Meier survival analysis, Breslow test: Chi square = 69.6, df = 3, p<0.001). Pairwise comparisons showed that, on average, fungus growth was first detected on honeybees that had not been embalmed and did not carry an egg, second were prey items that were embalmed but did not carry and egg, third were not embalmed honeybees carrying an egg and least susceptible were embalmed honeybees with an egg (Breslow test for all pairwise comparisons: Chi square ≥8.6, df = 1, p≤0.003). The timing of conidia formation followed the same pattern (Breslow test: Chi square = 67.4, df = 3, p<0.001; for all pairwise comparisons: Chi square ≥4.5, df = 1, p≤0.034; *Figure 8*).

## Synthesis of NO˙ in Beewolf eggs

Eukaryotes synthesize NO˙ from the amino acid L-arginine by the enzyme nitric oxide synthase (NOS) (*Röszer, 2012*) which is highly conserved also in insects (*Regulski and Tully, 1995*). The exceptional level of NO˙ emission of beewolf eggs raised the question of whether they employ the same pathway or have evolved a different mechanism. First, using histological staining, we assessed evidence for and site of NOS activity in beewolf eggs during the time of peak NO˙ emission. The fixation insensitive nicotinamide-adenine-dinucleotide phosphate (NADPH) -diaphorase assay resulted in strong blue staining only in embryonic tissue (*Figure 9*), thus indicating NOS activity. Second, by employing reverse transcription and real time quantitative PCR, we revealed that the temporal expression pattern of NOS-mRNA showed a clear peak around 19–20 hr after oviposition (*Figure 10*). For this experiment, the eggs were kept at 25°C, so the timing of peak NOS expression corresponds to the timing of peak NO˙ emission at this temperature (*Figure 6—figure supplement 1*). Third, to directly test for the involvement of NOS we injected beewolf eggs with Nω-nitro-L-arginine methylester (L-NAME), a NOS-inhibiting analog of L-arginine (*Willmot et al., 2005*). This treatment

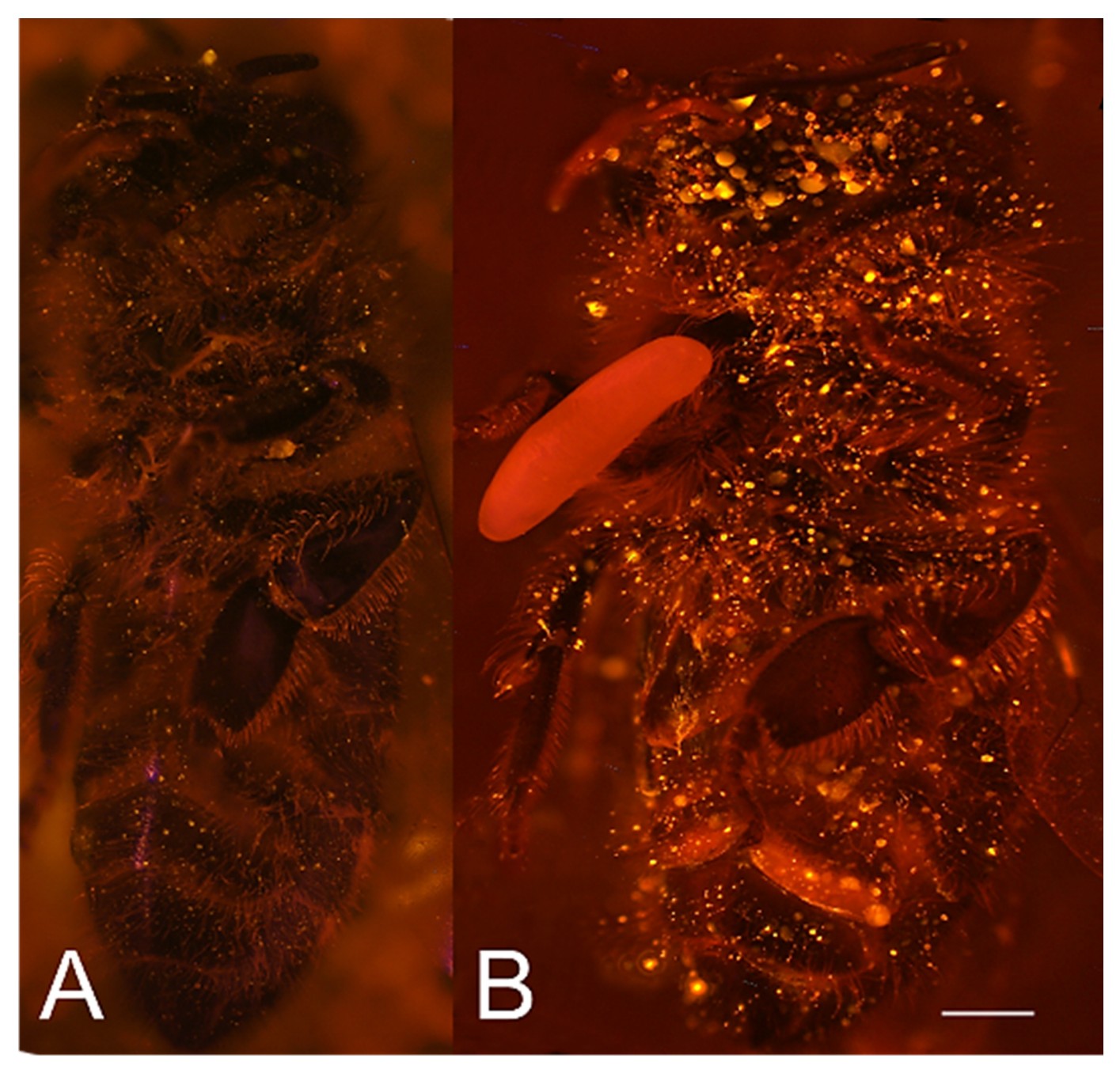

**Figure 4.** Visualization of NO˙ emission by beewolf eggs using fluorescence imaging. (**A**) Honeybee from a brood cell without an egg and (**B**) honeybee with egg. Both bees were sprayed with a solution of the NO˙ specific fluorescence probe DAR4M-AM. Only the droplets on the bee with the egg (**B**) show a bright yellow and orange fluorescence indicating the presence of NO˙. Images are composites of multiple pictures of the x/y plane and z-axis. Scale bar = 1 mm.

DOI: https://doi.org/10.7554/eLife.43718.007

The following figure supplement is available for figure 4:

**Figure supplement 1.** Results of the Griess assay with beewolf eggs.

DOI: https://doi.org/10.7554/eLife.43718.008

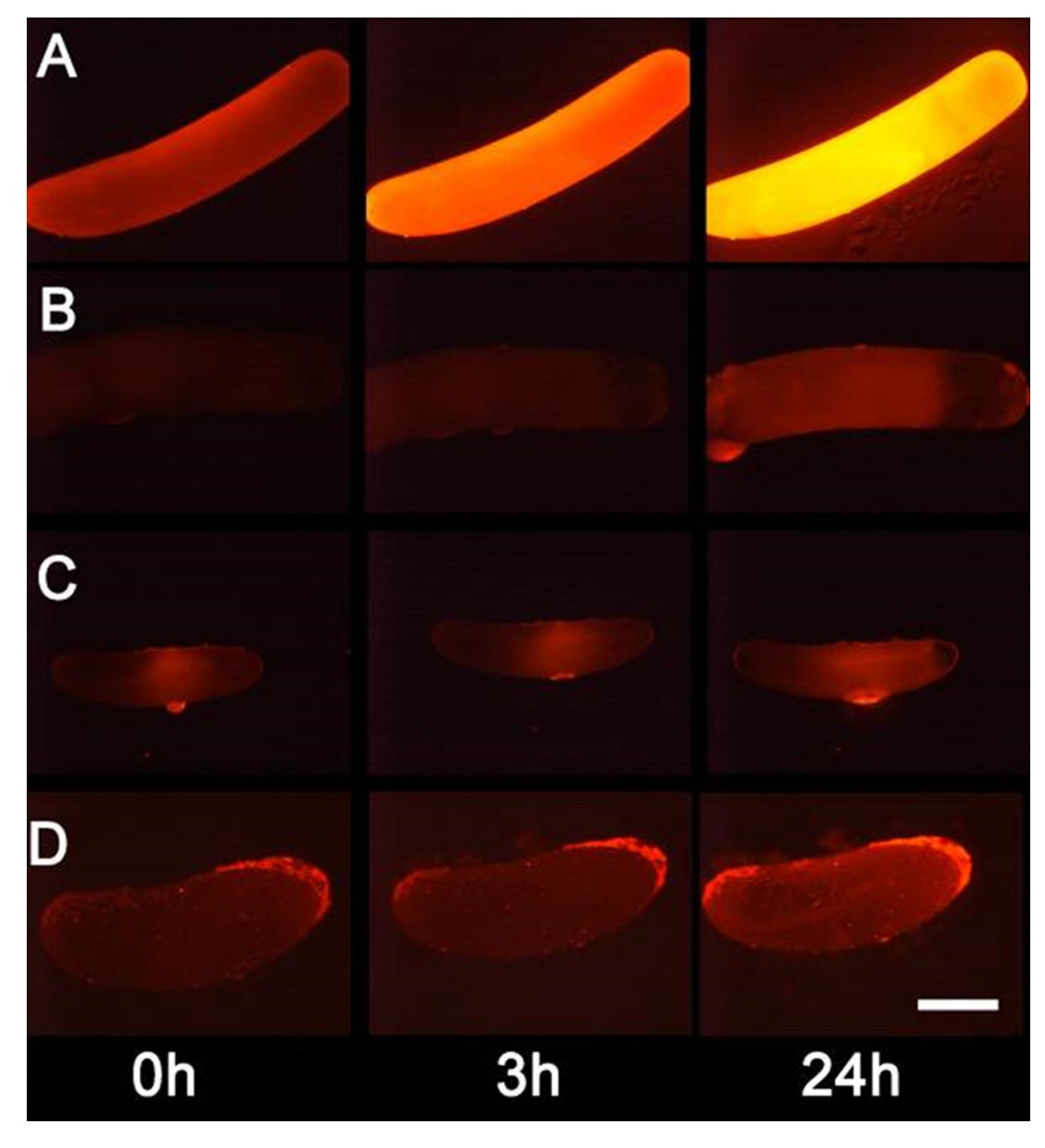

**Figure 5.** Detection of nitric oxide (NO˙) in beewolf eggs. Newly laid eggs of beewolves, *Philanthus triangulum*, of the cockroach wasp *Ampulex compressa* and of the Red Mason bee, *Osmia bicornis* were injected with the NO˙ sensitive fluorescence probe DAR4M-AM. Control beewolf eggs were injected with phosphate buffer. Images were obtained by fluorescence microscopy 0, 3 and 24 hr after injection. Row (**A**) DAR4M-AM injected beewolf egg showing strong increase in fluorescence; (**B**) Buffer-injected control beewolf egg showing the level of autofluorescence; (**C**) DAR4M-AM injected egg of *A. compressa;* (**D**) DAR4M-AM injected egg of *O. bicornis*. Scale bar: 1 mm.

DOI: https://doi.org/10.7554/eLife.43718.009

The following source data and figure supplement are available for figure 5:

**Source data 1.** Eggs injected with DAR4M-AM.
DOI: https://doi.org/10.7554/eLife.43718.011

**Figure supplement 1.** Detection of nitric oxide (NO˙) in beewolf larva.
DOI: https://doi.org/10.7554/eLife.43718.010

caused a significant decrease in NO˙ emission, whereas the non-inhibiting enantiomer D-NAME had no such effect (*Figure 11*).

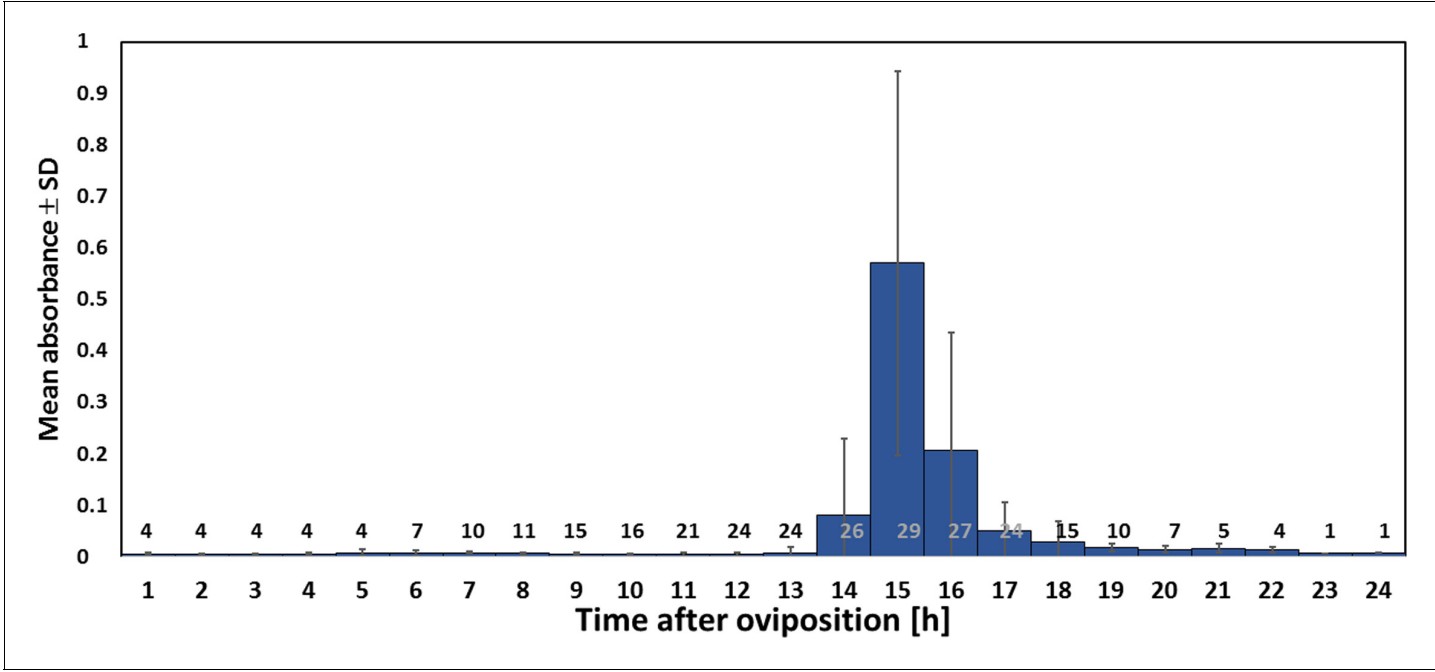

**Figure 6.** Timing of NO˙ emission from beewolf eggs (kept at 28˚C). The photometrically determined absorbance at 590 nm (mean ± SD) is shown as a function of time after oviposition for iodide-starch solutions successively exposed to beewolf eggs for one hour. Sample size (number of eggs measured) at each one hour interval is indicated above the x-axis.

DOI: https://doi.org/10.7554/eLife.43718.012

The following source data and figure supplements are available for figure 6:

**Source data 1.** Timing of NO emssion.

DOI: https://doi.org/10.7554/eLife.43718.014

**Figure supplement 1.** Start of NO˙ emission (h after oviposition) as a function of temperature.

DOI: https://doi.org/10.7554/eLife.43718.013

**Figure 6—figure supplement 1—source data 1.** Start of NO emission.

DOI: https://doi.org/10.7554/eLife.43718.015

## The beewolf *NOS*-gene and NOS-mRNA in eggs

In contrast to vertebrates, most invertebrates appear to have only one type of NOS (*Rivero, 2006*; *Whitten et al., 2007*). Considering the high level of NO˙ production in beewolf eggs, we hypothesized that beewolves have more than one *NOS* gene or that the NOS responsible for the NO˙ synthesis in beewolf eggs might exhibit considerable changes in enzyme structure compared to the NOS of related species. Sequencing of the *NOS*-gene(s) of *P. triangulum* (*Pt-NOS*) revealed only one *Pt-NOS* copy in the beewolf genome comprising 9.36 kbp with 25 exons (*Figure 11—figure supplement 1*). A phylogenetic analysis of the resulting amino acid sequence revealed a high similarity to the NOS of the closely related bees (Apidae, *Figure 11—figure supplement 2*). However, mRNA sequencing showed that, in contrast to adult beewolves and honeybees, the NOS-mRNA of beewolf eggs (3.72 kbp) lacks exon 14 comprising 144 bp. In the NOS-mRNA of adult beewolves this exon is located between the binding domains for calmodulin and flavin mononucleotide (FMN) (*Figure 11—figure supplement 1*).

## Discussion

Fighting pathogens is of outstanding importance for any organism and has driven the evolution of a great diversity of antimicrobial defenses. Internal immune systems have been extensively documented especially in vertebrates (*Akira et al., 2006*; *Hirano et al., 2011*) but also in insects (*Lemaitre and Hoffmann, 2007*; *Siva-Jothy et al., 2005*), including insect eggs (*Gorman et al., 2004*). However, comparatively little is known about external antimicrobial strategies that provide

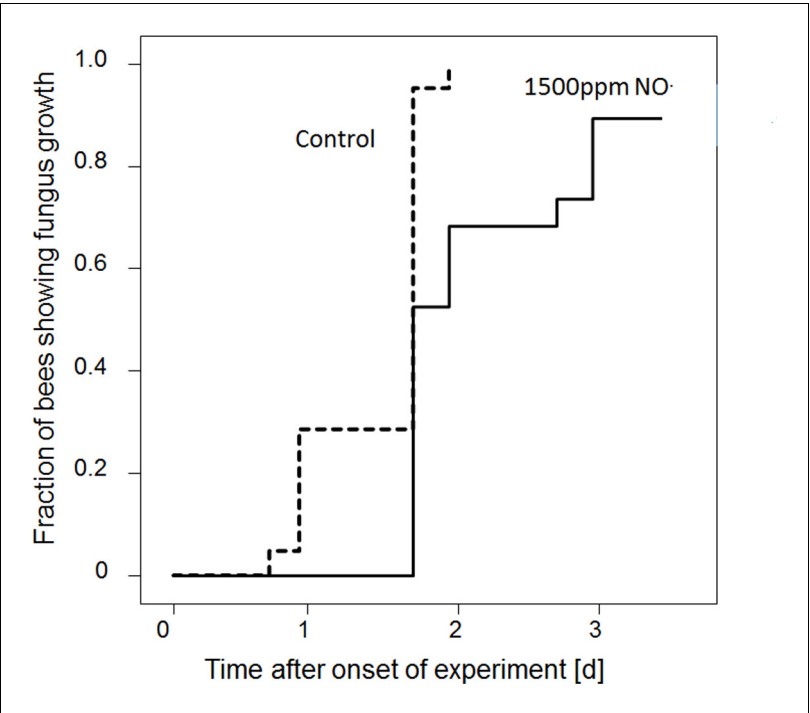

**Figure 7.** Onset of fungal growth (time after onset of experiment) on honeybees that were not embalmed in artificial brood cells. Brood cells were either injected with synthetic NO˙ to a concentration of 1500ppm (solid line) or were injected with nitrogen (dashed line) (N = 20 each, hazard ratio = 0.41, 95% confidence interval: 0.198–0.845).

DOI: https://doi.org/10.7554/eLife.43718.016

The following source data is available for figure 7:

**Source data 1.** Effect of synthetic nitric oxide on fungus growth.
DOI: https://doi.org/10.7554/eLife.43718.017

protection for the own body, for the progeny, or for food. Mechanical grooming is an important mechanism to remove microbes (*Zhukovskaya et al., 2013*). There are some reports on the application of antimicrobial secretions on the body surface by adult insects (*Wilson and Cotter, 2013*; *Otti et al., 2014*) or inside a host by larvae of a parasitoid wasp (*Herzner et al., 2013*). Carrion beetles preserve the larval food, buried carcasses, by application of antimicrobials (*Degenkolb et al., 2011*) and by controlling the microbiome on the carcasses (*Shukla et al., 2018*). Females of some insect species deposit antimicrobial chemicals (*Vander Meer and Morel, 1995*; *Marchini et al., 1997*) or antibiotics producing symbiotic bacteria (*Flórez et al., 2017*; *Flórez et al., 2015*) onto their eggs and ant workers can counter microbial infestation of the brood by applying venom (*Tragust et al., 2013*). Recently, the employment of volatile antimicrobials by insects as a means of external defense has gathered some interest (*Gross et al., 2008*; *Gross and Schmidtberg, 2009*; *Weiss et al., 2014*; *Lopes et al., 2015*).

Like other insects that develop in the soil, beewolves are particularly menaced by a diverse and unpredictable range of detrimental microbes. In fact, beewolf progeny and their provisions are under severe threat from fast growing mold fungi (*Strohm and Linsenmair, 2001*). The development of beewolf progeny from oviposition to cocoon spinning lasts about 11 days and is, thus, rather fast. So even a few days delay in fungus growth provides a considerable benefit for the larvae. Beewolves have evolved at least three very different antimicrobial defenses that provide an effective, coordinated, and long-term protection against a broad spectrum of microbes during the whole development. First, throughout the long period of winter diapause prior to emergence progeny are protected by antibiotics on their cocoons that are produced by symbiotic *Streptomyces* bacteria (*Kaltenpoth et al., 2005*; *Kroiss et al., 2010*; *Kaltenpoth et al., 2014*). Second, during the early egg and larval stages, molding of the provisions is retarded by an embalming of the honeybees with

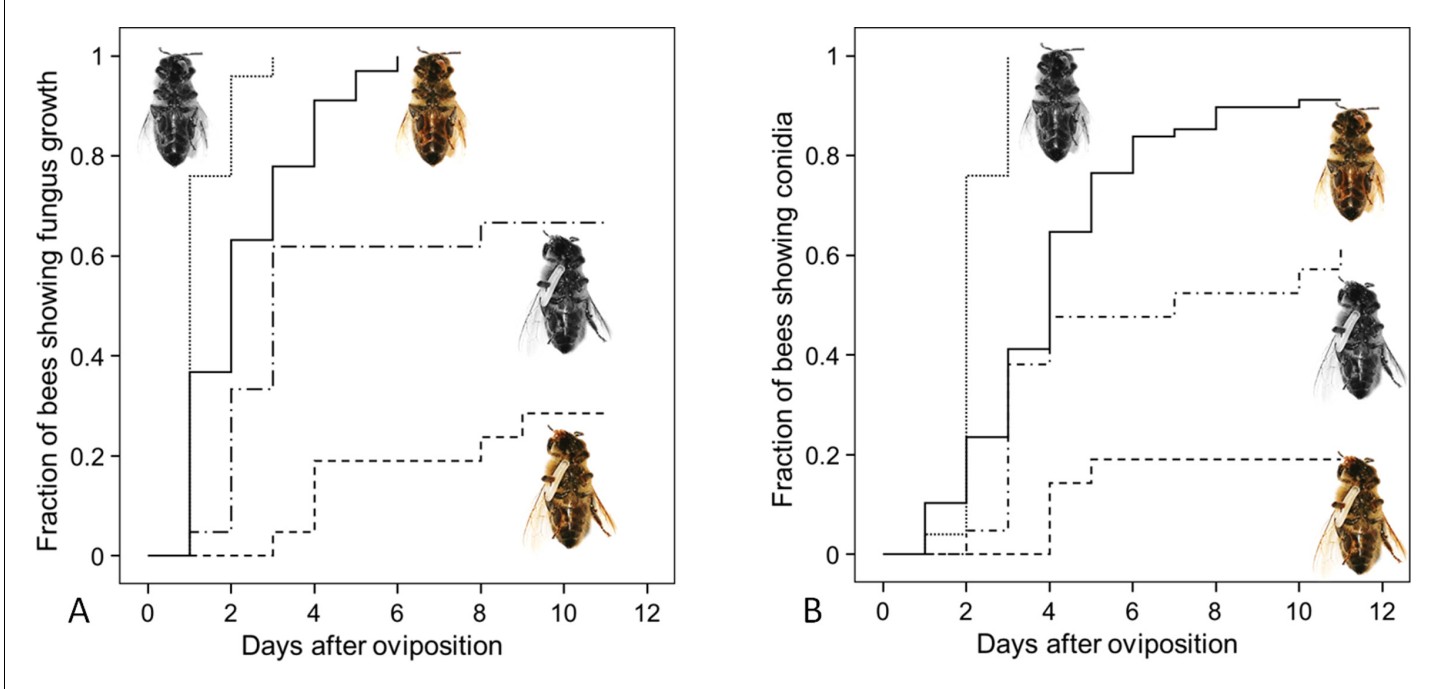

**Figure 8.** Fungus growth on honeybees of four different treament groups. Timing of occurrence of (**A**) fungal hyphae and (**B**) conidia on paralyzed honeybees that were (1) not embalmed by beewolf females and did not carry an egg (n = 25, colorless bee, point line), (2) embalmed but did not carry an egg (n = 68, colored bee, solid line), (3) not embalmed but carried an egg (n = 21, colorless bee with egg, dash-point line) or (4) embalmed and carried an egg (n = 21, colored bee with egg, dashed line). See *Appendix 1—table 2* for hazard ratios.

DOI: https://doi.org/10.7554/eLife.43718.018

The following source data is available for figure 8:

**Source data 1.** Combined effect of embalming and fumigation.

DOI: https://doi.org/10.7554/eLife.43718.019

lipids by the mother wasp (*Strohm and Linsenmair, 2001*). Third, as shown here, the emission of gaseous nitrogen oxide radicals by the beewolf egg results not only in delay of molding but in killing of detrimental fungi in their immediate environment thus, at least partly, eliminating this major threat.

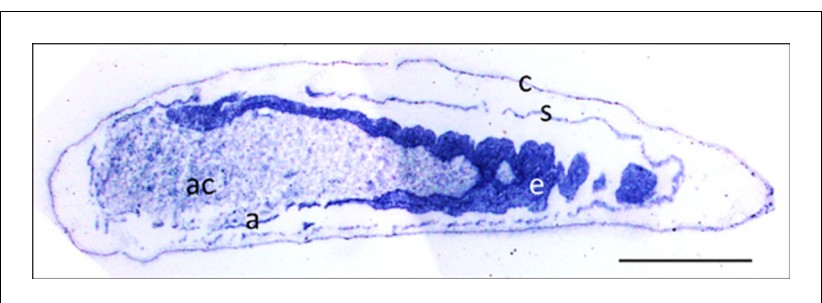

**Figure 9.** Micrograph of a longitudinal section of a beewolf egg fixed 15–16 hr after oviposition showing fixation insensitive NADPH-diaphorase activity. Strong blue staining in the embryonic tissue indicates the presence of reduced nitroblue tetrazolium demonstrating NOS activity (c = cuticle, s = serosa, e = embryo, a = amnion, ac = amnion cavity, scale bar = 1 mm, image composed from two separate photos of the left and right parts of the egg.).

DOI: https://doi.org/10.7554/eLife.43718.020

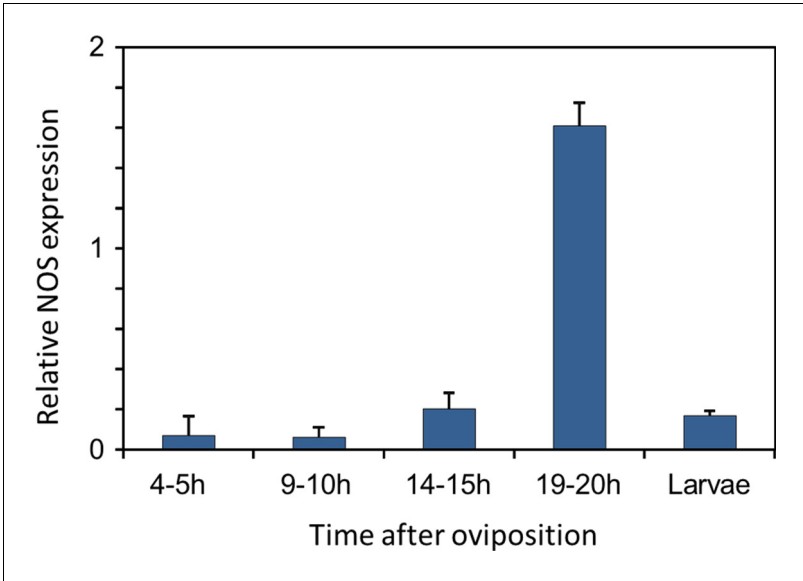

**Figure 10.** Gene expression of NOS relative to ß-actin in beewolf eggs at different times after oviposition and in freshly hatched larvae. Two trials were conducted, each with 25 pooled eggs or larvae per time interval. Mean ratios of NOS-mRNA to ß-Actin-mRNA are shown (with standard deviations), as determined by Q-RT-PCR.
DOI: https://doi.org/10.7554/eLife.43718.021

The following source data is available for figure 10:

**Source data 1.** NOS gene expression.
DOI: https://doi.org/10.7554/eLife.43718.022

---

The emission of a gaseous agent by beewolf eggs to their confined brood cells is an ideal way to sanitize such intricately structured surfaces as the bodies of honeybees and the rough walls of the brood cell. $NO^.$ seems to be a most suitable gaseous agent because it can obviously be produced by beewolf eggs in amounts that effectively kill mold fungi in their brood cell. Such volatile sanitation mechanisms that provide a front-line defense against microbes (*Gross et al., 2008*; *Weiss et al., 2014*; *Lopes et al., 2015*) will mostly be inconspicuous and might turn out to be a wider theme in nature.

Exact quantification of nitrogen oxides ($NO^.$ and $NO_2^.$) in beewolf brood cells on a micro scale or with time has not yet been accomplished. Brood cells are located in rather compact fine grained sandy soil with some moisture. Moreover, the walls of the nest burrows and the brood cells are covered with a layer of hydrocarbons (*Kroiss et al., 2009*) that might provide an additional barrier. Thus, brood cell walls are neither very porous nor are they sealed. Accordingly, the concentration of $NO^.$ and $NO_2^.$ in the brood cell will decrease but at a slow rate. By the time the larvae hatch (three days after oviposition) the smell of $NO^.$ has vanished, indicating that the nitrogen oxides have disappeared or at least decreased considerably, explaining why the larvae remain unaffected without actually being resistant to $NO^.$. However, even assuming some loss during the two hour period of peak $NO^.$ production the estimated maximum concentration of nitrogen oxides ($NO^.$ and $NO_2^.$) in beewolf brood cells (probably around 1500 ppm or 60 µmol/l) considerably exceeds the concentrations observed in animal tissues (mostly lower than 0.1 µmol/l [*Wink et al., 2011*], 0.85–1.3µmol/l in muscle tissue [*Vaughn et al., 1998*]). The maximum concentration in beewolf brood cells might be even higher than what is used in medical applications against multiple drug resistant bacteria (200 ppm $NO^.$[*Ghaffari et al., 2006*]) or in antifungal treatment of fruit (50–500 ppm $NO^.$[*Lazar et al., 2008*]) and is far beyond permissible exposure limits for humans (e.g. for the USA: 25 ppm for $NO^.$, 5 ppm for $NO_2^.$[*Administration USOSaH, 2014*]).

Synthetic $NO^.$ applied to artificial brood cells at a concentration of 1500ppm, the estimated concentration of nitrogen oxides in natural brood cells, significantly delayed fungus growth on bees. Since there was oxygen available in the brood cell, $NO^.$ was oxidized to $NO_2^.$ similarly to natural brood cells. The effect size ($NO^.$ treatment vs control, *Figure 7*: hazard ratio = 0.41, 95% confidence

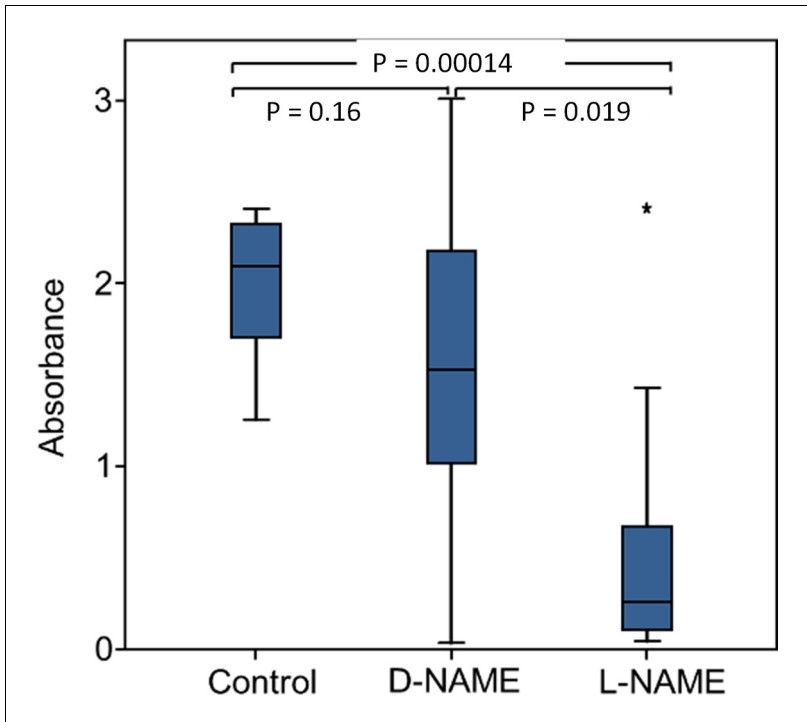

**Figure 11.** Effect of NOS inhibition on NO˙ production. Amount of NO˙ and/or NO$_2$˙ emanating from non-injected beewolf eggs (control; N = 14) and those injected with D-NAME (a non-inhibiting enantiomer of L-NAME, N = 9) or L-NAME (a NOS inhibiting L-arginine analog, N = 14). The photometrically determined absorbance at 590 nm is shown for iodide-starch solutions that were exposed for 24 hr to the headspace of eggs of the indicated treatment group (shown are median, quartiles and range, * indicates an outlier, included in the analysis). P-values are for Holm-corrected Mann-Whitney U-tests.

DOI: https://doi.org/10.7554/eLife.43718.023

The following source data and figure supplements are available for figure 11:

**Source data 1.** NOS inhibition.

DOI: https://doi.org/10.7554/eLife.43718.026

**Figure supplement 1.** Structure of the *Pt-NOS* gene indicating position and length of exons.

DOI: https://doi.org/10.7554/eLife.43718.024

**Figure supplement 2.** Consensus tree obtained from Bayesian analysis of NOS amino acid sequences from five orders of insects (distinguished by different colors), including the NOS sequences of *P. triangulum* eggs (lowermost entry).

DOI: https://doi.org/10.7554/eLife.43718.025

interval 0.198–0.845) was slightly lower than in a comparable experiment with eggs as the source of the antifungal gas (data for unembalmed bees from the experiment of a combined effect of embalming and emissions from the egg: *Appendix 1—table 2*: hazard ratio = 0.22, 95% confidence interval 0.1–0.47). However, since the confidence intervals of the hazard ratios are mutually overlapping, there is no evidence for a significant difference in the antifungal effect of 1500ppm synthetic NO˙ and the gas emitted by beewolf eggs. Although we cannot exclude that small amounts of other active volatiles are released by beewolf eggs, we thus conclude that the antifungal effect of brood cell fumigation by beewolf eggs is predominantly or exclusively due to NO˙ and its oxidation product NO$_2$˙.

Notably, the combination of prey embalming with unsaturated hydrocarbons that reduces condensation of water on the bees (*Herzner and Strohm, 2007*) and brood cell fumigation with NO˙/ NO$_2$˙ seems to affect fungal growth beyond either of these antimicrobial measures alone. One possible explanation is based on the fact that NO˙ and NO$_2$˙ dissolve in water (confirmed by the spraying of a bee with a fluorescent dye, *Figure 4*) to yield nitric acid and nitrous acid, with the latter being a potent antibacterial agent (*Gao et al., 2015*). Although embalming reduces the amount of water

condensation on the prey bees, some very small droplets occur. The concentration of nitrous acid in these droplets will be considerably higher than in the larger droplets that would occur without embalming. Thus, fungal germs that might have survived the $NO^{.}/NO_2^{.}$ atmosphere are not only impaired by limited availability of water, but the accessible water might be toxic for them. Moreover, due to the solubility of $NO^{.}$ and $NO_2^{.}$, abundant water droplets on the bees could reduce the concentration of these radicals in the brood cell, thus lessening their antimicrobial effect. The reduction of water on the bees due to prey embalming could thus help to keep fumigation effective. The combination of prey embalming and fumigation, thus, has a twofold effect. Many fungi will be killed by the $NO^{.}/NO_2^{.}$. The remaining spores will encounter unfavorable conditions that slow-down germination and growth so that the majority of larvae are able to consume most of their provisions without severe competition by mold fungi. Once larvae have spun their cocoon the antibiotics that are produced by the symbiotic bacteria take over protection until emergence (*Kaltenpoth et al., 2005*; *Kroiss et al., 2010*; *Engl et al., 2018*). Within this multifaceted antimicrobial strategy of beewolves, brood cell fumigation might be the most important component since it takes effect at a very early developmental stage and, thus, provides beewolf offspring with a decisive head start over the fast growing mold fungi.

NO$^{.}$ is an ancient biological effector of immense importance for all kinds of organisms ranging from prokaryotes to higher plants and animals (*Röszer, 2012*; *Moroz and Kohn, 2007*). Owing to its high diffusibility across biomembranes and specific chemical properties, this gaseous radical plays a crucial role in a multitude of biological processes (*Röszer, 2012*; *Moroz and Kohn, 2007*). In vertebrates, NO$^{.}$ is synthesized from L-arginine by three different isoforms of NOS that are encoded by different genes (*Röszer, 2012*; *Moroz and Kohn, 2007*). Low levels of NO (<1µmol/l) are produced by constitutive NOS (cNOS) isoforms (endothelial eNOS, neuronal nNOS) and have signaling functions, for example in neuronal development and in the regulation of vascular tone in vertebrates. Higher NO$^{.}$ concentrations (1–10 µmol/l, [*Thomas et al., 2003*]) are generated by an inducible NOS (iNOS). At such levels NO$^{.}$ is highly cytotoxic (*Thomas et al., 2003*), making it a powerful antimicrobial (*Fang, 1997*; *Lai et al., 2011*), for example in macrophages (*Röszer, 2012*). However, overproduction of NO$^{.}$ due to inflammatory processes (*Filipović et al., 2010*) or certain diseases (e.g. Alzheimer's disease, [*Lüth et al., 2002*]) may cause harmful side-effects (*Pacher et al., 2007*) and even septic shock (*Titheradge, 1999*). Moreover, NO$^{.}$ might affect carcinogenesis and tumor progression in a positive as well as in a negative way (*Burke et al., 2013*).

In living tissues, NO$^{.}$ is usually removed within seconds by reacting with the heme group of molecules such as oxyhemoglobin (*Beckman and Koppenol, 1996*; *Wink et al., 2011*) (very low concentrations may still persist for hours [*Moroz and Kohn, 2007*]). In brood cells, there is enough oxygen (670 µl) to support the metabolism of the egg and of the paralyzed bee as well as the oxidation of NO$^{.}$ (for more details see Appendix 1: Additional discussion 1). In air, the autooxidation to $NO_2^{.}$ is comparatively slow so that NO$^{.}$ may persist (depending on its concentration) for several seconds to minutes (*Mur et al., 2011*; *Wink et al., 2011*) or even hours (*Soegiarto et al., 2003*). Thus, the NO$^{.}$ emitted by beewolf eggs might directly affect fungi, for exapmle by damaging DNA (*Lai et al., 2011*; *Jones et al., 2010*) or by reacting with the heme group of enzymes like cytochrome P450 and cytochrome c oxidase, thus inhibiting these crucial components of the mitochondrial respiratory chain (*Thomas et al., 2003*; *Feelisch, 2008*; *Canessa and Larrondo, 2013*). Yet, most of the antimicrobial activity of NO$^{.}$ is attributed to indirect effects via reactive nitrogen species (RNS), in particular nitrogen oxides ($NO_2^{.}$, $N_2O_3$) and peroxynitrite ($ONOO^{-}$, upon reaction with superoxide) (*Thomas et al., 2003*). $NO_2^{.}$, has been reported to be severely cytotoxic, for example by nitration of tyrosine residues and oxidation of proteins and lipids (*Fang, 1997*; *Bogdan, 2001*).

A beewolf egg of approximately 5 mg emits 0.25 µmol NO$^{.}$ within a period of about 2.5 hr, or 20.000 µmol/kg*h, a value that is about four orders of magnitude higher than reported baseline levels of NO$^{.}$ synthesis in humans (0.15 - ~ 4.5 µmol/kg*h [*Castillo et al., 1996*], rats (0.6–9 µmol/kg*h [*Wu et al., 1999*]) and plants (*Arabidopsis thaliana*, 0.36–3 µmol/kg*h [*Zeidler et al., 2004*]), and even considerably higher than in lipopolysaccharide (LPS)-activated macrophages (~800 µmol/kg*h, estimated from *Wu et al., 1999*). To investigate whether a NOS was involved in NO$^{.}$ production in beewolf eggs we conducted three experiments. First, a specific histochemical assay indicated NOS activity in embryonic tissue but not in other parts of the egg. Second, quantitative PCR revealed elevated expression of the NOS gene at the time of peak NO$^{.}$ production. Finally, competitive inhibition of NOS by L-NAME caused a significant reduction in NO$^{.}$ production. While each of the three

results might have alternative explanations (e.g. L-NAME might not be a perfectly specific NOS inhibitor [*Peterson et al., 1992*]), taken together these findings provide strong evidence that a NOS, located in the embryonic tissue, is involved in NO˙ production of beewolf eggs.

Searching for possible adaptations that might accomplish this extremely high rate of NO˙ production the beewolf NOS gene was sequenced. Only one beewolf NOS (*Pt-NOS*) gene was found. The derived amino acid sequence did not reveal considerable differences compared to the NOS of the closely related bees (*Figure 11—figure supplement 2*). Thus, there is no evidence for extensive evolutionary changes with regard to the gene itself. Moreover, the structure of the *Pt-NOS* gene is largely homologous to other insects, for example *Anopheles stephensi* mosquitoes (*Luckhart et al., 1998*).

However, in contrast to adult beewolves, the NOS-mRNA in beewolf eggs lacks exon 14 (144 bp, *Figure 11—figure supplement 1*). Such alternative splicing that results in different NOS-mRNAs, including the deletion of exons (but others than in beewolves), has been documented in *A. stephensi* in response to *Plasmodium* infection (*Luckhart and Li, 2001*). Moreover, NOS splice variants may result in organ-specific enzymes in other organisms (*Röszer, 2012*). Presumably, beewolf eggs produce smaller amounts of another NOS splice variant to support signaling functions in the developing embryo.

In adult beewolves, the exon missing in the NOS-mRNA of eggs is located between the binding domains for calmodulin and FMN. Since calmodulin is believed to be responsible for NOS regulation (*Smith et al., 2013*) the deletion of an adjacent part might affect the control of NOS activity in beewolf eggs. Thus, the alternative splicing might enable the production of such large amounts of NO˙. Notably, compared to the cNOS (comprising eNOS, and nNOS) the inducible NOS isoform of vertebrates (iNOS) that generates higher concentrations of NO˙ to combat microbes lacks a section of about 40 amino acids (120 bp) near the FMN domain. Interestingly, this section is thought to be responsible for autoinhibition of the cNOS (*Salerno et al., 1997*) and its lack enhances NO˙ production by the iNOS. The conspicuous similarity between vertebrate iNOS and the NOS in beewolf eggs with regard to the length of the missing section and its position might suggest a convergent modification to achieve a NOS with high synthetic capacity. Whereas vertebrates have evolved another gene, beewolf eggs might accomplish a similar effect by alternative splicing of the mRNA.

The possible loss of regulation of the NOS and the pattern of *Pt-NOS* expression in the eggs suggest that in beewolf eggs the activity of the enzyme is regulated by gene expression like the NOS in *Plasmodium* infested *A. stephensi* (*Luckhart et al., 1998*) and the iNOS of vertebrates (*Wong et al., 1996*; *Morris Jr, 1999*). However, in contrast to these caes, in beewolf eggs expression of the *Pt-NOS* seems not to be induced by immunostimulants but to occur obligatorily at a certain stage in the development of the beewolf embryo. While we cannot exclude that there is an additional, yet unknown, pathway of NO˙ production in beewolf eggs, we hypothesize that the NOS and in particular its alternative splice variant plays a significant role in brood cell fumigation by beewolf eggs.

However, even the combined effect of prey embalming and brood cell fumigation does not provide perfect protection as fungus infestation still causes larval mortality in 5% of the brood cells in the field (*Strohm and Linsenmair, 2001*). Some fungal spores might survive under the bees because they were screened against the gas. Another possibility, namely that strains of the ubiquitous mold fungi that are the main causes of molding in beewolf brood cells (*Engl et al., 2016*), have evolved resistance against the toxic effects of NO˙/NO$_2$˙ seems rather unlikely. Ultimately, there will be only weak selection for resistance at all since beewolf brood cells are certainly a rare habitat for the ubiquitous mold fungi (*Engl et al., 2018*). Moreover, there will be no repeated exposure of the same fungal strains to fumigation that would be required to favor the evolution of resistance. While there are examples for detoxification of lower concentrations of NO˙ (mainly by scavengers like flavohemoglobins) in different fungi, including species of *Aspergillus* (*Martins et al., 2011*; *Zhou et al., 2009*), the NO˙/NO$_2$˙ levels emitted by beewolf eggs are very high and likely affect several very basic biochemical processes, thus making the evolution of an effective resistance unlikely.

While brood cell fumigation clearly retards molding of larval provisions, the antimicrobial effect of NO˙ and NO$_2$˙ might harm the symbiotic *Streptomyces* bacteria that beewolf females apply to the brood cell prior to egg laying (*Kaltenpoth et al., 2005*; *Kroiss et al., 2010*). Since the symbiotic bacteria are important for the survival of larvae in the cocoon and are vertically transmitted from beewolf mothers to their daughters (*Kaltenpoth et al., 2014*), a considerable number of symbionts have to survive the brood cell fumigation. At the moment we can only speculate how the bacteria

can survive. Conceivably, because of strong selection due to specialization and repeated exposition, the symbiotic bacteria have evolved mechanisms to cope with the high concentrations of NO·/NO$_2$· (*Poole, 2005*; *Wareham et al., 2018*). Possibly, the fumigation slowly evolved after the establishment of the symbiosis; thus bacteria might have been able to gradually evolve resistance. Moreover, the bacteria are applied to the ceiling of the brood cell, which might reduce negative effects of the nitrogen oxides since these are heavier than air (*Lide, 1995*) and will accumulate in the lower part of the brood cell. Additionally, the bacteria are embedded in copious amounts of a secretion consisting of mostly unsaturated hydrocarbons (*Kaltenpoth et al., 2009*) that might shield the bacteria from the fumigants. Finally, host- and/or symbiont derived antioxidants in the hydrocarbon matrix could detoxify NO· and NO$_2$· and protect the symbiotic *Streptomyces* bacteria.

How could brood cell fumigation with high concentrations of NO·/NO$_2$· have evolved? Generally, it has been assumed that the primary purpose of NO· was signaling at low concentrations and that the antimicrobial functions of higher concentrations are derived (*Fang, 2004*). Assuming a similar scenario for beewolves, small amounts of NO· that were originally produced for developmental processes (*Andersen et al., 2013*) might have accidentally been released into the confines of the subterranean brood cell and slightly affected the germination or growth of fungi by interfering with regulatory processes (*Röszer, 2012*; *Wang and Higgins, 2005*). Given the severe threat posed by microbes, such initial benefits would have caused strong selection for elevated NO· emission by the eggs. This would have considerably increased progeny survival and might have allowed ancestral beewolves to nest in an expanded range of habitat types, including nesting sites with high risk of microbial infestation, or to exploit highly susceptible but readily available prey species. Brood cell fumigation with large doses of NO· thus represents a key evolutionary innovation. Since NO· is used as an antimicrobial in the immune systems of many animals (*Bogdan et al., 2000*), its deployment as an antifungal gas can be viewed as an innate, externalized immune defense of beewolf eggs. Such externalized components of the immune system have recently been recognized as important and possibly widespread antimicrobial measures (*Otti et al., 2014*).

The clear benefit of brood cell fumigation, however, is probably accompanied by substantial costs in terms of energy and biochemical resources (*Rivero, 2006*). NO· is synthesized from L-arginine, an amino acid that is an important constituent of many proteins and biochemical pathways (*Morris Jr, 2000*) and it is an essential amino acid for most insects (*Barbehenn et al., 1999*; *Payne and Loomis, 2006*) (e.g. phytophagous insects [*Berenbaum, 1995*], mosquitos [*Uchida, 1993*], aphids [*Sasaki and Ishikawa, 1995*; *Akman Gündüz and Douglas, 2009*], butterflies [*Erhardt and Rusterholz, 1998*; *O'Brien et al., 2003*], true bugs [*Mesquita et al., 2015*], parasitoid wasps [*Thompson, 1976*; *Barrett and Schmidt, 1991*], bees [*de Groot, 1952*; *Weiner et al., 2010*]). Thus, beewolves have either evolved the capacity to synthesize L-arginine or female beewolves have to provide each egg with sufficient L-arginine for both brood cell fumigation and embryogenesis. Moreover, NO· synthesis by NOS requires the cofactors flavin adenine dinucleotide (FAD), FMN, (6R-)5,6,7,8-tetrahydrobiopterin (BH4) and NADPH (*Förstermann and Sessa, 2012*), thus competing with other metabolic pathways in the developing beewolf embryo.

One of the most remarkable aspects of our study is that the embryos inside the egg survive the high concentrations of toxic nitrogen oxides during synthesis and emission as well as after its release to the brood cell. This is all the more surprising since beewolf larvae that were accidentally exposed to the gas emitted by eggs died (Strohm, unpublished observations). The synthesis and emission of such high amounts of NO· likely requires a number of concomitant adaptations that protect beewolf embryos against the cytotoxic effects of high concentrations of NO· and NO$_2$·. One possibility is the employment of carrier molecules to transfer NO· to the egg shell. In blood sucking hemipterans, for example, nitrophorins carry NO· to its release site to dilate blood vessels (*Davies, 2000*). The mechanistic basis of NO· tolerance of beewolf eggs is of particular interest, since excessive production of NO· due to inflammatory processes (*Guzik et al., 2003*) or certain diseases (e.g. Alzheimer's disease, [*Lüth et al., 2002*; *Pacher et al., 2007*; *Calabrese et al., 2007*; *Pautz et al., 2010*]) might cause severe pathological complications in humans. Thus, understanding how beewolf eggs avoid the toxic effects of NO· might inspire the development of novel medical applications.

Our findings reveal a surprising adaptation in a mass-provisioning digger wasp to cope with the threat of pathogen infestation in the vulnerable egg and larval stages. Sanitizing the brood cell environment by producing high amounts of NO· significantly enhances the survival of immatures by reducing fungal growth on their provisions. Given that mass-provisioning and development

underground are widespread ecological features among digger wasps and bees and considering the difficulties of detecting volatiles in subterranean nests, such gaseous defenses might be more widespread and as yet underappreciated. In addition to revealing new perspectives on antimicrobial strategies in nature and amplifying the biological significance of NO., beewolves offer unique opportunities to elucidate general questions on the evolution and regulation of NOS as well as the production of and resistance to high concentrations of NO..

# Materials and methods

## Key resources table

| Reagent type (species) or resource | Designation | Source or reference | Identifiers | Additional information |
|---|---|---|---|---|
| Biological sample | European beewolf, *Philanthus triangulum* | Field caught or laboratory reared F1 of field caught females | | |
| Biological sample | Emerald cockroach wasp, *Ampulex compressa* | Laboratory reared | | |
| Biological sample | Red mason bee, *Osmia bicornis* | Field caught | | |
| Biological sample | *Aspergillus flavus* | Strain I: Isolated from beewolf brood cells, Strain II: Department of Hygiene and Microbiology of the University Hospital, Würzburg, Germany | na | |
| Biological sample | *Penicillium roquefortii* | Department of Hygiene and Microbiology of the University Hospital, Würzburg, Germany | na | |
| Biological sample | *Candida albicans* | Department of Hygiene and Microbiology of the University Hospital, Würzburg, Germany | na | |
| Biological sample | *Trichophyton rubrum* | Department of Hygiene and Microbiology of the University Hospital, Würzburg, Germany | na | |
| Sequence-based reagent | Adapter + PolyT | 3'RACE, Molecular cloning protocol | | See *Supplementary file 1* |
| Sequence-based reagent | Adapter | 3'RACE, Molecular cloning protocol | | See *Supplementary file 1* |
| Sequence-based reagent | polyT | Reverse transcription protocol | | See *Supplementary file 1* |
| Sequence-based reagent | NOS_qPCR_F2 | *P. triangulum*, this paper | | See *Supplementary file 1* |
| Sequence-based reagent | NOS_qPCR_R2 | *P. triangulum*, this paper | | See *Supplementary file 1* |
| Sequence-based reagent | Actin_qPCR_F1 | *Apis mellifera, Gryllus bimaculatus, P. triangulum*, this paper | | See *Supplementary file 1* |
| Sequence-based reagent | Actin_qPCR_R1 | *A. mellifera, G. bimaculatus, P. triangulum*, this paper | | See *Supplementary file 1* |

*Continued on next page*

*Continued*

| Reagent type (species) or resource | Designation | Source or reference | Identifiers | Additional information |
|---|---|---|---|---|
| Sequence-based reagent | NOS860fwd2 | *A. mellifera, D. melanogaster, Anopheles stephensi, Rhodnius prolixus, Manduca sexta,* this paper | | See *Supplementary file 1* |
| Sequence-based reagent | NOS1571rev1 | *A. mellifera, D. melanogaster, A. stephensi, R. prolixus, M. sexta,* this paper | | See *Supplementary file 1* |
| Sequence-based reagent | NOS_seq_F1_deg | *A. mellifera, Nasonia vitripennis,* this paper | | See *Supplementary file 1* |
| Sequence-based reagent | NOS_seq_R1_deg | *A. mellifera, N. vitripennis,* this paper | | See *Supplementary file 1* |
| Sequence-based reagent | NOS_seq_5-F1 | *P. triangulum,* this paper | | See *Supplementary file 1* |
| Sequence-based reagent | NOS_seq_5-R1 | *P. triangulum,* this paper | | See *Supplementary file 1* |
| Sequence-based reagent | NOS_seq_5-F2 | *P. triangulum,* this paper | | See *Supplementary file 1* |
| Sequence-based reagent | NOS_seq_5-R2 | *P. triangulum,* this paper | | See *Supplementary file 1* |
| Sequence-based reagent | NOS_seq_5-F3 | *P. triangulum,* this paper | | See *Supplementary file 1* |
| Sequence-based reagent | NOS_seq_5-F6 | *P. triangulum,* this paper | | See *Supplementary file 1* |
| Sequence-based reagent | NOS_seq_3-F1 | *P. triangulum,* this paper | | See *Supplementary file 1* |
| Sequence-based reagent | NOS_seq_3-R1 | *P. triangulum,* this paper | | See *Supplementary file 1* |
| Sequence-based reagent | NOS_seq_3-F2 | *P. triangulum,* this paper | | See *Supplementary file 1* |
| Sequence-based reagent | NOS_seq_3-R2 | *P. triangulum,* this paper | | See *Supplementary file 1* |
| Sequence-based reagent | NOS_seq_3-F3 | *P. triangulum,* this paper | | See *Supplementary file 1* |
| Sequence-based reagent | NOS_seq_3-F6 | *P. triangulum,* this paper | | See *Supplementary file 1* |
| Sequence-based reagent | NOS_RT_R1 | *P. triangulum,* this paper | | See *Supplementary file 1* |
| Commercial assay or kit | Griess assay. Merck Spectroquant | Merck, Darmstadt, Germany | 114776 | |

*Continued on next page*

*Continued*

| Reagent type (species) or resource | Designation | Source or reference | Identifiers | Additional information |
|---|---|---|---|---|
| Commercial assay or kit | peqGOLD total RNA Kit | peqLab, Erlangen, Germany | 732–2867 | |
| Commercial assay or kit | GeneRacer Kit | Invitrogen, Carlsbad, CA, USA | L1502-01 | |
| Commercial assay or kit | BioScript One-Step RT-PCR-Kit | Bioline, London, UK | BIO-65033 | |
| Commercial assay or kit | peqGOLD Taq-DNA-Polymerase | peqLab, Erlangen, Germany | 01–1030 | |
| Commercial assay or kit | SensiMixPlus SYBR Mit | Quantace/Bioline, London, UK | QT615-05 | |
| Commercial assay or kit | Epicentre MasterPure Complete DNA and RNA purification Kit | Epicentre, now Lucigen, Middleton, WI, USA | MC85200 | |
| Commercial assay or kit | innuPREP RNA Mini Kit | Analytik Jena, Jena, Germany | 845-KS-2040050 | |
| Commercial assay or kit | PeqGOLD Mid-Range PCR System | peqLab, Erlangen, Germany | PEQL02-3020_P | |
| Chemical compound, drug | DNase I | Fermentas, Lithuania Now Thermo Fisher Scientific, Germany | EN0525 | |
| Chemical compound, drug | Oligo-dT primer | Fermentas, Lithuania Now Thermo Fisher Scientific, Germany | na | |
| Chemical compound, drug | L-NAME Hydrochloride | Axxora Deutschland, Lörrach, Germany | ALX-105–004 M250 | |
| Chemical compound, drug | D-NAME Hydrochloride | Axxora Deutschland, Lörrach, Germany | ALX-105–003 G005 | |
| Chemical compound, drug | DAR-4M AM | Axxora Deutschland, Lörrach, Germany | ALX-620–069 M001 | |
| Chemical compound, drug | 4-Nitro-m-Xylol | Merck, Darmstadt, Germany | 8415470025 | |
| Chemical compound, drug | NADPH-Tetranatriumsalz | Carl-Roth, Karlsruhe, Germany | AE14.1 | |
| Tools | Eppendorf Microinjector with Femtotips II | Eppendorf, Hamburg, Germany | 930000043 | |
| Tools | Axiophot II Fluorescence microscope | Zeiss, Jena, Germany | | |
| Tools | Nikon DS-2 Mv | Nikon, Tokyo, Japan | | |
| Tools | Uvikon 860 spektrophotometer | Kontron, Augsburg, Germany | | |
| Tools | Cryostat microtome CM3000 | Leica, Wetzlar, Germany | | |
| Tools | Eppendorf Realplex Cycler | Eppendorf, Hamburg, Germany | | |

*Continued on next page*

*Continued*

| Reagent type (species) or resource | Designation | Source or reference | Identifiers | Additional information |
|---|---|---|---|---|
| Tools | NanoDrop TM1000 | peqLab, Erlangen, Germany | RRID: SCR_016517 | |
| Tools | Implen Nanophotometer Classic | Implen, Munich, Germany | | |
| Tools | Biometra T Gradient Thermocycler | Analytik Jena, Jena, Germany | | |
| Software | BioEdit | http://www.mbio.ncsu.edu/BioEdit/bioedit.html | RRID: SCR_007361 | |
| Software | Geneious | Biomatters, New Zealand | | |
| Software | SPSS | IBM, Armonk, NY, USA | RRID: SCR_002865 | |
| Software | FastTree | http://www.microbesonline.org/fasttree/ | RRID: SCR_015501 | |
| Software | MrBayes | http://mrbayes.sourceforge.net/ | RRID: SCR_012067 | |
| Software | Combine-ZP | www.hadleyweb.pwp.blueyonder.co.uk | | |
| Software | Photoshop Elements 5 | PSE5, Adobe Systems Inc, San José, CA, USA | | |
| Software | CLC genomics workbench | Qiagen, Hilden, Germany | RRID: SCR_011853 | |
| Database | NCBI | http://www.ncbi.nlm.nih.gov | RRID: SCR_006472 | |
| Database | Primer3 | http://primer3.ut.ee | RRID: SCR_003139 | |
| Dervice | Sanger Sequencing | Seqlab, Göttingen, Germany | | |
| Service | Transcriptome Sequencing on Illumina HiSeq TM2000 | Fasteris, Geneva, Switzerland | | |

## Animals

Beewolf females, *Philanthus triangulum* F. (Apoidea, Crabronidae), were either caught in the field from populations in Franconia (Germany) or were the F1 progeny of such females kept in the laboratory. They were housed in observation cages (*Strohm and Linsenmair, 1994*) that provided access to newly completed brood cells. The cages were placed in a room with temperature control (20–22° at night, 25–28°C in the daytime) and were lit for 14 hr per day by neon lamps. Honeybees, *Apis mellifera* L. (Apoidea, Apidae), the females' prey, were caught from hive entrances or from flowers and provided *ad libitum*. Honey was provided *ad libitum* in the flight cage for the nutrition of both honeybees and beewolf females.

To obtain freshly laid eggs, observation cages were checked hourly. Completed brood cells were opened, their length and width was measured using calipers and the egg and/or honeybees were removed and used for the experiments. Brood cell volume was estimated as a prolate spheroid with brood cell length as the major and width as the minor axis. The bees in brood cells had been paralyzed, embalmed with lipids (*Herzner et al., 2007*), and provisioned by beewolf females. Egg volume was estimated by calculating the volume of a cylinder with the respective length and width of an egg (both determined using a stereomicroscope with eyepiece micrometer). The temperatures at which eggs were kept reflect natural conditions (*Herzner and Strohm, 2008*) and allow for an optimal development (*Strohm, 2000*).

## General experimental procedures

For all experiments beewolf eggs were harvested from brood cells of various females. Eggs were randomly allocated to different treatment groups. Sample sizes refer to independent biological replicates, that is each replicate represents a different egg or brood cell – with the exception of quantitative PCR, where several eggs were pooled for one sample (see below). As it is very demanding to obtain beewolf eggs, the availability of eggs of a certain developmental stage was limited. Generally, we used as many eggs as feasible (e.g. for quantitative PCR). For some experiments we decided on a meaningful sample size based on experience from preliminary experiments (e.g. we already knew that inhibition assays with beewolf eggs in Petri dishes were really clear-cut and required only few replicates). Moreover, due to the limited availability of beewolf eggs on a given day, replicates were conducted consecutively over several days.

## Fungus inhibition assays

To test whether the time course of fungus growth on bees differed between those carrying an egg and those without egg, we used brood cells (N = 22) that had been provisioned with two bees. We placed each bee individually into an artificial brood cell of natural shape and volume in sand-filled Petri dishes (diameter 10 cm) and with moisture levels similar to natural conditions. Petri dishes were placed in a climate chamber at 25°C in the dark. Bees were carefully checked visually every 24 hr for fungus growth without opening the Petri dishes. First signs of fungus infestation (hyphae) were recorded. The experiment was terminated after eleven days since all larvae had finished feeding and spun a cocoon by then. Since these are time event data, we used survival analysis (Kaplan Meier, Breslowe test; hazard ratios and their 95% confidence intervals are presented as estimates of effect sizes, SPSS Statistics 24) to compare the timing of fungus infestation of the bees with and without an egg. Larvae hatched on the third day after oviposition and started to feed on the bee. There was no evidence that hatched larvae were able to prevent fungus growth on the bee they occupied or others in the brood cell. However, to take a possible effect of the larva on the experimental bee into account, we carried out the analysis not only over the whole period from oviposition until the larvae spun into a cocoon (11 days) but also for the period from oviposition to the hatching of larvae (3 days). A significant difference already until the third day indicates that this effect was associated with the egg.

We examined whether beewolf eggs emit a volatile antimicrobial by conducting two experiments. For the first test, we used brood cells (N = 16) that contained three bees. The bees were transferred to artificial brood cells in sand-filled Petri dishes as described above. The bee with the egg and one of the bees without egg (the experimental bee) were placed together in the same artificial brood cell but without physical contact. The other bee without egg (the control) was kept alone in another artificial brood cell (in another Petri dish). We monitored the timing of fungus infestation as described above. We used survival analysis as described above. Again, to take an (unlikely) effect of the larva into account, we also carried out the analysis for the period from oviposition until the larvae hatched (day 3 after oviposition). A significant difference already until day three could only be caused by volatiles emanating from the egg.

For the second assay, we exposed conidiospores of a diverse spectrum of fungi to the volatiles emanating from beewolf eggs. Petri dishes (10 cm) containing culture medium (malt extract agar or Sabouraud-agar [*Atlas, 2004*]) were inoculated with conidia from different fungal strains (*Aspergillus flavus* strain A, Trichocomaceae, that was isolated from infested beewolf brood cells, [*Engl et al., 2016*], N = 20; *A. flavus* strain B, *Mucor circinelloides*, Mucoraceae; *Penicillium roquefortii*, Trichocomaceae; *Candida albicans*, Saccharomycetaceae; *Trichophyton rubrum*, Arthrodermataceae; N = 8 for all the latter strains and species; these were kindly provided by the Department of Hygiene and Microbiology of the Würzburg University Hospital). Conidiospores were harvested by sampling mature fungus colonies that were reared from stock cultures. A suspension of the conidia in sterile water was evenly distributed on the Petri dishes to obtain uniform growth of fungi. To recreate the concentrations of potential antibiotic volatiles in the brood cell, we used small plastic caps (3 ml, about the size of a brood cell) to confine test areas on the agar. Freshly laid eggs were placed singly on the bottom of a cap where they readily attached due to their natural stickiness. Each cap was then placed on a freshly inoculated Petri dish so that the agar under the cap was not in contact with the egg but was exposed to volatiles that emanated from the egg. An empty cap was placed on the same Petri dish as a control. The Petri dishes were incubated in a dark climate chamber at 25°C.

Fungus growth under the experimental and control caps was recorded after 24, 48 and 72 hr. After 72 hr the caps with the hatched larvae were removed, and fungal growth was further recorded after another 24, 48 and 72 hr. Since the results were clear-cut with either no fungal growth or substantial growth (*Figure 3*) and no intermediate cases, the experimental and control areas were compared using binomial tests (software PAST [*Hammer et al., 2001*]).

## Identification of the antimicrobial volatile

We hypothesized that nitric oxide (NO˙) and its main reaction product with oxygen, nitrogen dioxide (NO$_2$˙), were the most likely compounds emanating from beewolf eggs. The standard test for the detection of NO˙ and NO$_2$˙ employs the Griess reaction. We used a solution of sulfanilic acid and N-(1-naphthyl)-ethylenediamine (Spectroquant Nitrite Test, Merck, Germany, according to the manufacturer's instructions). The Griess reagent specifically reacts with the nitrite anion (NO$_2^-$) to form a distinctive red azo dye (*Guevara et al., 1998*). NO˙ reacts with water to form nitrous acid (HNO$_2$) and can thus be directly verified by the Griess reaction. NO$_2$˙, however, disproportionates in water into nitrous acid and nitric acid (HNO$_3$) and the latter must be reduced to nitrous acid to react with the Griess reagent. Freshly laid beewolf eggs (collected within 2 hr after oviposition, N = 11) were placed in the lid of a 1.5 ml reaction tube where they readily attached due to their natural stickiness. Tubes without eggs (N = 11) were used as controls. Then 1 mL of the Griess test solution was added to the tube. For another sample (N = 15) the nitrate, which might be present in the solution, was reduced to nitrite by placing a glass fiber filter disc with small amounts of zinc powder (*Jander and Blasius, 1971*) on the surface of the solution. The same setting without an egg was used as control (N = 15). The tubes were incubated at 25°C for 24 hr, and the occurrence of the red coloration was examined visually and with a photometer (at 520 nm, Nanophotometer, Implen, Germany, quantitative measurements were not meaningful with this set-up since the azo dye is not perfectly stable over time, according to the manufacturer's instructions). The samples with and without nitrate reduction showed qualitatively the same results.

NO˙ can also be detected by specific fluorescent probes. In particular, diaminorhodamin-4M AM (DAR4M-AM), a cell permeable, photostable fluorescent dye, has a high sensitivity and specificity for NO˙[*Kojima et al., 2000*]). A DAR4M-AM (Alexis Biochemicals, USA) solution was prepared according to the supplier's instructions (10µmol/l in 0.1 mol/l phosphate buffer, pH 7.4). To verify and to visualize the emission of NO˙ from the egg, paralyzed honeybees either with freshly laid eggs (N = 8) or controls without eggs (N = 8) were sprayed with the DAR4M-AM solution using a nebulizer (the egg itself was screened from droplets during spraying) and kept in the dark (at 25°C in artificial brood cells as described above). After 20 hr, the bees were examined under a fluorescence microscope (Axiophot II, Zeiss, Germany, filter set 43: excitation 520–570 nm, emission 535–675 nm) and digital photos were taken (Nikon DS-2 Mv, Nikon Japan) at constant exposure times, to allow comparison of fluorescence intensity. Due to the size of the bees, several pictures had to be taken in the X,Y plane as well as along the Z axis. Pictures along the z-axis were stacked using the software Combine-ZP (www.hadleyweb.pwp.blueyonder.co.uk). Then these stacks were stitched using Photoshop Elements 5 (PSE5, Adobe Systems Inc USA). Since small peripheral background parts within the frame of the stacked and stitched picture were 'empty' these parts were filled with other background parts by using the clone stamp tool. Images were corrected for contrast and sharpness using PSE5 with identical settings for experimental and control specimens.

DAR4M-AM can also be used to detect NO˙ in tissues. Aliquots of 0.1–0.5 µl of the DAR4M-AM solution (see above) were injected into beewolf eggs (within 1 hr after oviposition, N = 64, in N = 45 eggs the embryo survived and developed) with a custom made microinjector equipped with glass capillaries (Eppendorf Femtotips II, Eppendorf, Germany) under microscopic control. Control eggs injected with buffer only (N = 10) were monitored in the same way to assess autofluorescence. For comparison, eggs of two other Hymenoptera (*Osmia bicornis*, Apoidea, Megachilidae, N = 12, and *Ampulex compressa*, Apoidea, Ampulicidae, N = 9; eggs from both species were obtained from our own laboratory populations) as well as freshly hatched beewolf larvae (N = 4) were injected with the DAR4M-AM solution. All eggs were kept in dark chambers at 25°C (a temperature within the optimal range for development for all these species), and fluorescence was observed directly after injection and 1, 3, 5, 24, sometimes 48 and 72 hr later. For some eggs not all time points were available. Fluorescence was examined under a fluorescence microscope and documented with a digital camera as

described above. Contrast and sharpness of the images were optimized using Photoshop Elements 5 (Adobe, USA) with identical settings for all specimens.

## Iodometry: Quantification, time course and temperature dependence of NO˙ production

Iodometry provides a simple but sensitive, reliable and precise method to quantify strong oxidants. To assess the amount of emitted nitrogen oxides, we placed freshly laid eggs (N = 233) individually into the lid of 1.5 ml reaction tubes where they readily attached due to their natural stickiness. Then 1 ml of a potassium iodide-starch solution (containing 1% KI and 1% soluble starch in distilled water) was added, the reaction tube was closed and kept for 24 hr at 28°C in a dark climate chamber. Oxidation of iodide results in iodine that forms a blue complex with starch (*Jander and Blasius, 1971*). The degree of coloration was quantified by measuring the absorbance at 590 nm in a spectrophotometer (Uvikon 860, Kontron, Germany). To assess the absolute amount of the oxidant, the solutions were subsequently calibrated by titration with a reference solution of sodium thiosulfate (concentration: 0.001 M; Merck, Germany) until the blue color of the iodine-starch complex disappeared.

To establish the time course of gas production, individual beewolf eggs (N = 4) were transferred within 1 hr after oviposition into the lid of reaction tubes and kept in a dark climate chamber at 28°C. Every hour, the cap with the egg was transferred to another reaction tube with fresh iodide-starch solution. Immediately after removal of the egg from a reaction tube, absorbance of the solution was measured at 590 nm as described above.

To investigate the temperature dependence of gas production, tubes with a newly laid egg and iodide-starch solution (as described above, N = 33 in total) were placed in a rack (with white background) inside a climate chamber and incubated at seven different constant temperatures (20, 22.5, 24, 25.5, 27, 28.5°C and 30°C). The time course of coloration of the iodide-starch solution was recorded using a digital camera (Canon EOS 20D, Canon, Japan) programmed to take pictures at 30 min intervals. The onset of gas production could be easily determined since the color of the solution turned from clear to dark blue from one picture to the next, that is within a 30 min interval. A quadratic regression curve was fitted to the data (SPSS Statistics 24) and the $Q_{10}$ value for the temperature dependence was estimated.

## Bioassay to test for the antifungal effect of synthetic NO˙

We assessed the effect of synthetic NO˙ on the beginning of fungus growth on honeybees in artificial brood cells. Sand filled Petri dishes with artificial brood cells (volume 3 ml) were prepared as described above and a honeybee (collected at the entrance of a bee hive and killed by freezing) was placed into the brood cell. Nitric oxide was generated by the oxidation of zinc powder with nitric acid ($HNO_3$) (*Jander and Blasius, 1971*). In order not to affect the composition of the gases in the artificial brood cell, we adjusted the concentration of the generated NO˙ so that an addition of 10% of the volume of the brood cell resulted in an initial concentration of 1500ppm NO˙. Employing iodometry as described above, we adjusted the amounts of reactants so that the addition of 300 μl to the brood cell volume of 3 ml resulted in a NO˙ concentration of 1500ppm (the presumable peak concentration in natural brood cells). Zinc powder (0.5 g) was placed in a vial (20 ml) and the vial was closed with a plastic lid with two small holes (~0.5 mm, one for pressure compensation). Then the vial was extensively flushed with pure nitrogen to remove oxygen that would otherwise oxidize NO˙ to $NO_2$˙. Immediately, 150 μL of 20% $HNO_3$ were added with an insulin syringe so that the resulting gas mixture in the vial was composed of 15000ppm NO˙ and 98.5% $N_2$. Using a gastight syringe (Hamilton, Reno, NV, USA) an aliquot of 300 μl of this gas mixture was injected into the artificial brood cell with a bee through a small hole (~0.5 mm) in the lid of the Petri dish and the hole was immediately closed with adhesive tape (N = 20). Thus the concentration of NO˙ in the artificial brood cell was 1500ppm. As controls, otherwise identically prepared Petri dishes with bees in artificial brood cells were injected with 300 μl of pure nitrogen (N = 20).

The Petri dishes were incubated in a dark climate chamber at 25°C. All bees were carefully checked for fungus growth under a stereomicroscope for three days (twice per day). As a consequence, first signs of fungus growth were detected earlier than in other experiments of this study, where we used the unaided eye. Data were analysed using survival analysis as described above. To assess whether synthetic NO˙ has a similar antifungal effect as the gas emitted by beewolf eggs we

compared the effect size (hazard ratio) of this experiment with the data testing for the effect of the egg produced gas on unembalmed bees (as part of the experiment on the 'combined effects', see below). If the 95% confidence intervals of the hazard ratios overlap, there is no evidence for a difference between the effects.

## Bioassay to test for a combined effect of NO˙ and embalming

Fungus growth and conidia formation on bees of four different experimental groups were recorded for eleven days (until the larvae had spun their cocoons). The groups consisted of: (1) paralyzed honeybees that were not embalmed and did not carry an egg (n = 25), (2) paralyzed and embalmed honeybees without egg (n = 68), (3) paralyzed honeybees that were not embalmed but an egg was carefully transferred onto them from another bee (n = 21), and (4) paralyzed and embalmed honeybees with egg (n = 21). To control for effects of the transfer of eggs in group (3), each egg in group (4) was sham treated by using tweezers to lift it up from the bee and putting them back onto the same bee. Non-embalmed bees were removed from beewolf females immediately after paralysation. Embalmed bees were removed from brood cells in observation cages within 12 hr after oviposition. All bees were transferred to artificial brood cells (one bee per brood cell) in Petri dishes filled with moist sand. The Petri dishes were incubated in a dark climate chamber at 25°C. All bees were checked daily for both fungus growth and formation of conidia under a stereomicroscope. As above, first signs of fungus growth were detected earlier than in the experiments of this study, where we used the unaided eye. Data were analysed using survival analysis as described above with pairwise comparisons of treatment groups (SPSS 24).

## Detection of NOS activity in egg tissue

To assess whether there was NOS activity in the egg tissue and where it was located, we used fixation-insensitive NADPH diaphorase staining with nitroblue tetrazolium (*Virgili et al., 2001*; *Müller, 1994*). Eggs were fixed in PBS containing 4% paraformaldehyde for 2 hr at 4°C, followed by cryoprotection in PBS with 12% sucrose for 20 hr. The tissue was soaked in Tissue Tec (Sakura Finetek, Netherlands) for 30 min, frozen, and 10 µm sections were cut on a cryostat microtome (CM3000, Leica, Germany). The sections were incubated for 60 min at 30°C with 50 mmol/l Tris-HCI, pH 7.8, 0.1% Triton X-100, and 0.2 mmol/l nitroblue tetrazolium chloride in the presence or absence (each N = 5) of 0.2 mmol/l β-NADPH to demonstrate fixation-insensitive NADPH diaphorase activity. The sections were dehydrated, mounted with Depex (Serva, Germany) and observed under a compound microscope (Zeiss Axiophot II). Photos were taken with a digital camera (Nikon DS-2 Mv). Since the egg was larger than the field of view of the camera, two pictures had to be taken and were stitched (Photoshop Elements 5, Adobe USA). Contrast and sharpness were optimized.

## Phenology of NOS gene expression

If NOS is responsible for NO˙ production in beewolf eggs, the time pattern of *NOS* gene expression should largely resemble the time course of NO˙ production by showing a pronounced peak several hours after egg laying (the timing of the peak depending on temperature). We used reverse transcription and real time quantitative PCR to quantify the NOS mRNA in beewolf eggs at different times after oviposition. Since the amount of mRNA that could be obtained from single eggs was insufficient to get reproducible results, we conducted two trials for each of four different time intervals after oviposition (4–5, 9–10, 14–15 and 19–20 hr after oviposition). For each trial and per each time interval we pooled 25 eggs (all kept at 25°C), as well as 25 freshly hatched larvae repsectively. The eggs and larvae were removed from the brood cells at the specified times, shock frozen with liquid nitrogen and stored at −80°C. The RNA of each sample was extracted using the peqGOLD total RNA Kit (Peqlab, Germany) according to the supplier's instructions and eluted with 20 µL RNase free water. An aliquot of 3 µL of the RNA was digested with DNaseI (Fermentas, Lithuania) and transcribed into cDNA with BioScript (Bioline, Germany) using an Oligo-dT primer (Fermentas, Lithuania) in a final volume of 20 µL. As a reference for basic levels of gene expression during the experimental period, mRNA of the housekeeping gene β-actin was quantified and the ratio of NOS/β-actin mRNA was calculated for each sample.

For quantitative PCR, we established new primers for both the NOS and β-actin genes of *P. triangulum* (based on the complete *NOS* sequences, see below) (NOS_qPCR_F1 and R4; Actin_qPCR_F1

and R1, *Supplementary file 1*). All primers were intron-overlapping to avoid the measurement of contaminating genomic DNA. The NOS and actin primers amplified fragments of 312 bp and 321 bp, respectively. The specificity of both primer sets was confirmed by sequencing purified PCR products. The qPCRs were performed on an Eppendorf Realplex cycler (Eppendorf, Germany) in a final volume of 25 µL, containing 1 µL of template cDNA (1 µL of the 20 µL RT reaction mix), 2.5 µL of each primer (10 pmol/l) and 12.5 µL of SYBR Green Mix (SensiMixPlus SYBR Mit, Quantace, UK). Standard curves were established by using $10^{-9}$ – $10^{-3}$ ng of PCR products as template. A Nano-Drop TM1000 spectrophotometer (Peqlab, Germany) was used to measure DNA concentrations of the templates for the standard curves. PCR conditions were as follows: 95°C for 5 min, followed by 50 cycles of 56°C (β-actin) or 65°C (NOS) for 60 s, 72°C for 60 s and 95°C for 60 s. Then a melting curve analysis was performed by increasing the temperature from 60°C to 95°C within 20 min. Based on the standard curves, the amount of NOS and β-actin template and their ratio was calculated.

## NOS inhibition assay

To verify the role of NOS in NO˙ production by beewolf eggs, we used an inhibition assay (*Willmot et al., 2005*). Since L-arginine is the substrate for NO˙ production by NOS, we injected either an inhibiting L-arginine analog or, for controls, a non-inhibiting enantiomer into freshly laid beewolf eggs. Chemicals were dissolved in 0.1 mol/l phosphate buffer pH 7.4. Using a microinjector (see above) eggs were injected with about 0.2 µl of 1.5 mol/l solutions of (1) the competitive inhibitor Nω-nitro-L-arginine methylester (L-NAME, Sigma-Aldrich, USA) (experimental group, N = 14), or (2) the non-inhibiting Nω-nitro-D-arginine methylester (D-NAME, Sigma-Aldrich, USA) (control group 1, N = 9) or (3) not injected at all (N = 14, control group 2). Each egg of the three groups was placed individually in the lid of a reaction tube with an iodide-starch solution as described above and incubated for 24 hr at 28°C. Then NO˙ production was assessed by measuring absorbance of the solution with a photometer (Implen Nanophotometer) at 590 nm. Statistical comparison of the groups was conducted using Mann-Whitney U-tests with correction after Holm (*Holm, 1979*) (SPSS Statistics 24).

## Sequencing of the *P. triangulum* nitric oxide synthase gene (*Pt-NOS*) and mRNA

DNA was extracted from female beewolf heads with the Epicentre MasterPure Complete DNA and RNA Purification kit (Epicentre, USA) according to the manufacturer's guidelines for tissue extraction. Eggs for RNA extraction were kept at a temperature of 27.5°C (range 26–29°C), collected 14–15 hr after oviposition, immediately frozen in liquid nitrogen and stored at −70°C until RNA extraction. Twenty eggs were pooled for extraction and homogenized by repeatedly pipetting in lysis buffer of the PeqGOLD Total RNA kit (Peqlab, Germany). Samples were processed according to the kit manual and frozen at −70°C. For the full transcriptome sequencing (to obtain the 5' terminal region) RNA was extracted from the antennae of eight frozen female beewolves according to manufacturer's protocol 1 of the innuPrep RNA Mini Kit (Analytik Jena, Germany).

Most of the beewolf *NOS* gene was amplified and sequenced by primer walking. Sequencing reactions were performed by a commercial service (Seqlab, Germany). Four degenerate primers (NOS860fwd2, NOS1571rev1, NOS_seq_F1_deg, and NOS_seq_R1_deg) were designed (*Supplementary file 1*) based on published *NOS* sequences of *Drosophila melanogaster* (U25117.1), *Apis mellifera* (AB204558.1), *Anopheles stephensi* (AH007775.1), *Rhodnius prolixus* (U59389.1), *Manduca sexta* (AF062749.1) and *Nasonia vitripennis* (NM_001168232.1). First, the central region (~700 bp, between NOS860fwd2 and NOS1571rev1) was amplified and sequenced. Based on this sequence, we designed a pair of *P. triangulum* specific primers (NOS_qPCR_F2 and NOS_qPCR_R2, *Supplementary file 1*). Using one specific central and one degenerate terminal primer (NOS_seq_F1_deg and NOS_seq_R1_deg, *Supplementary file 1*), respectively, fragments of 4–5 kb were amplified and sequenced by primer walking, which yielded the central 9.5 kb of the *NOS* gene.

Fragments larger than 2 kb were amplified with the PeqGOLD Mid-Range PCR System on a thermocycler (TGradient, Biometra, Germany). Reaction volumes of 12.5 µL contained 1 µL DNA template, 50 mmol/l Tris-HCl (pH 9.1), 14 mmol/l $(NH_4)_2SO_4$, 1.75 mmol/l $MgCl_2$, 350 mmol/l of each dNTP, 400 mmol/l of each primer and 0.5 U 'MidRange PCR' enzyme mix. An initial 3 min melting step at 94°C was followed by 35 cycles of 0.5 min at 94°C, 0.5 min at 58°C and 3 min +20 s per cycle at 68°C and a final extension time of 20 min at 68°C.

Fragments up to 2 kb were amplified using the PeqGOLD Taq. Reaction volumes of 12.5 µL contained 1 µL of DNA template, 50 mmol/l Tris-HCl pH 9.1, 14 mmol/l (NH$_4$)$_2$SO$_4$, 3 mmol/l MgCl$_2$, 240 µmol of each dNTP, 800 nmol/l of each primer and 0.5 U Taq. An initial 3 min melting step at 95°C was followed by 35 cycles of 1 min at 95°C, 1 min at 60°C and 2 min at 72°C and a final extension time of 3 min at 72°C.

The 3' terminus was sequenced following the 3' RACE protocol (*Sambrook and Russell, 2001a*). Briefly, cDNA was generated by reverse transcription with a poly-T primer. Before reverse transcription, co-extracted DNA was digested using DNaseI (New England Biolabs, UK).The DNA digestion mix contained 1 mmol/l Tris-HCl, 0.25 mmol/l MgCl$_2$ and 1 mmol/l CaCl$_2$ and 0.4 U DNaseI. DNA was digested for 10 min at 37°C, followed by DNase inactivation for 10 min at 75°C. The final reverse transcription mix contained 25 mmol/l KCL, 10 mmol/l Tris-HCl, 0,6 mmol/l MgCl$_2$, 2 mmol/l DTT, 4 µmol poly-T or gene specific primer, 0.5 mmol/l of each dNTP and 200 U of BioSkript Moloney Murine Leukemia Virus reverse transcriptase (Bioline, Germany). The entire digestion mixture was incubated with the primer for 5 min at 70°C to enable primer annealing, then cooled on ice. Reverse transcription was carried out for 1 hr at 42°C and the enzyme was subsequently inactivated for 10 min at 70°C. The cDNA including the 3' terminal region was amplified with the specific primer NOS_-seq_3-F3 and a 'poly-T adapter primer', that is a polyT primer to which a specific adapter sequence was added (*Sambrook and Russell, 2001b*) (*Supplementary file 1*). Subsequently, a nested PCR was performed using a second specific primer (NOS_seq_3-F6) and a primer that contained only the specific adapter sequence of the 'polyT adapter primer' to increase PCR specificity (*Supplementary file 1*, same PCR conditions as above).

The 5' terminal region of 200 bp was obtained from a full transcriptome sequencing approach of female antennae, which covered the full-length NOS mRNA sequence. RNA sequencing was performed by a commercial service provider (Fasteris, Switzerland), using the HiSeq TM2000 Sequencing System (Illumina, USA) with 100 bp single reads, on 5 µg total RNA isolated from female *P. triangulum* antennae. CLC Genomics Workbench was used for sequence assembly of the resulting 75 million reads. Reads were quality-trimmed with standard settings and subsequently assembled using the following CLC parameters: nucleotide mismatch cost = 2; insertion cost = 2; deletion cost = 2; length fraction = 0.3; similarity = 0.9. Conflicts among the individual bases were resolved by voting for the base with highest frequency. Contigs shorter than 250 bp were discarded.

To sequence the entire NOS transcript from eggs, cDNA was generated by reverse transcription with a poly-T primer and additionally a specific, central NOS_RT_R1 primer, followed by PCR amplification using various primer combinations to cover the whole transcript sequence (*Supplementary file 1*). Additionally, the sequence of the 5' terminal region was confirmed by RT-PCR of mRNA from *P. triangulum* eggs, using primers NOS_seq_5-F6 and NOS_seq_5-R3 (*Supplementary file 1*) and subsequent sequencing.

Even though we used a large number of primers to cover the gene, we did not find sections with signals for two different bases at the same site. Thus we infer that there is only one *NOS* gene in the *P. triangulum* genome, as in most invertebrates (*Labbé et al., 2009*). In addition, the transcriptome dataset did not reveal any other transcript that was annotated as nitric oxide synthase.

The GenBank accession numbers for the *P. triangulum NOS* (*Pt-NOS*) gene sequence is: KJ425525, for the NOS mRNA of *P. triangulum* eggs: KJ425526, and for the NOS mRNA in *P. triangulum* female antennae: KJ425527.

## Phylogenetic analysis of *NOS* gene sequences

NOS coding sequences of 23 insect species from five orders were acquired from the NCBI database. Along with the *P. triangulum NOS* sequence, these were translated and aligned using Geneious (Version 6.0.5, created by Biomatters, Geneious, New Zealand). The highly variable 5' end was trimmed. An approximately-maximum-likelihood tree was created with FastTree (*Price et al., 2010*; *Price et al., 2009*). Local support values were estimated with the Shimodaira-Hasegawa test based on 1000 samples without re-optimizing the branch lengths for the resampled alignments (*Price et al., 2010*). Bayesian estimates were made with the program MrBayes 3.1.2 (*Huelsenbeck et al., 2001*; *Huelsenbeck and Ronquist, 2001*; *Ronquist and Huelsenbeck, 2003*). The MCMC analysis was conducted under a mixed amino acid rate model (prset aamodelpr = mixed). After 1,000,000 generations, with trees sampled every 1000 generations, the standard deviation of split frequencies was consistently lower than 0.01. We discarded the first 100

of the sampled trees (10% burn-in) and computed a 50% majority rule consensus tree with posterior probability values for every node. The trees estimated by both methods were nearly identical, so they were combined into a single figure.

## Acknowledgements

We are grateful to Wilhelm Boland for advice, to Clarissa Schill and Nathalie Moske for technical assistance, to Stephan Schneuwly for providing the microinjection device and, in particular, to Jon Seger, Jeremy Field and to the anonymous reviewers for valuable comments on earlier drafts of the manuscript.

## Additional information

### Funding
No external funding was received for this study.

### Author contributions
Erhard Strohm, Conceptualization, Resources, Data curation, Formal analysis, Supervision, Validation, Investigation, Visualization, Methodology, Writing—original draft, Project administration, Writing—review and editing; Gudrun Herzner, Resources, Investigation, Methodology, Writing—review and editing; Joachim Ruther, Conceptualization, Investigation, Methodology, Writing—review and editing; Martin Kaltenpoth, Conceptualization, Resources, Investigation, Methodology, Writing—review and editing; Tobias Engl, Conceptualization, Formal analysis, Investigation, Visualization, Methodology, Writing—review and editing

### Author ORCIDs
Erhard Strohm https://orcid.org/0000-0002-8899-9765
Martin Kaltenpoth http://orcid.org/0000-0001-9450-0345
Tobias Engl https://orcid.org/0000-0002-2200-2678

### Decision letter and Author response
Decision letter https://doi.org/10.7554/eLife.43718.055
Author response https://doi.org/10.7554/eLife.43718.056

## Additional files

### Supplementary files
• Supplementary file 1. Primers used for sequencing of the *Pt-NOS*.
DOI: https://doi.org/10.7554/eLife.43718.027

• Transparent reporting form
DOI: https://doi.org/10.7554/eLife.43718.028

### Data availability
The GenBank accession numbers for the P. triangulum NOS (Pt-NOS) gene sequence is: KJ425525, for the NOS mRNA of P. triangulum eggs: KJ425526, for the NOS mRNA in P. triangulum female antennae: KJ425527, for the transcriptome: Bioproject accession number PRJNA542283 with bio-sample accession number SAMN11793601. Other data generated or analysed during this study are included in the manuscript and supporting files. Source data files have been provided for Figures 2, 5, 6, 6—supplement 1, 7, 8, 10, 11.

The following datasets were generated:

| Author(s) | Year | Dataset title | Dataset URL | Database and Identifier |
|---|---|---|---|---|
| Strohm E, Herzner G, Ruther J, Kaltenpoth M, Engl T | 2015 | Philanthus triangulum nitric oxide synthase gene, complete cds, alternatively spliced | https://www.ncbi.nlm.nih.gov/nuccore/KJ425525 | GenBank, KJ425525.1 |

| | | | | |
|---|---|---|---|---|
| Strohm E, Herzner G, Ruther J, Kaltenpoth M, Engl T | 2015 | Philanthus triangulum nitric oxide synthase egg isoform mRNA, complete cds, alternatively spliced | https://www.ncbi.nlm.nih.gov/nuccore/KJ425526 | GenBank, KJ425526.1 |
| Strohm E, Herzner G, Ruther J, Kaltenpoth M, Engl T | 2015 | Philanthus triangulum nitric oxide synthase adult isoform mRNA, complete cds, alternatively spliced | https://www.ncbi.nlm.nih.gov/nuccore/KJ425527 | GenBank, KJ425527.1 |
| Sandoval-Calderón M, Nechitaylo T, Engl T, Vogel H, Koehler S, Kaltenpoth M | 2019 | RNAseq reads from Philanthus triangulum and its symbiont, 'Candidatus Streptomyces philanthi' | https://www.ncbi.nlm.nih.gov/bioproject/542283 | Genbank, PRJNA542283 |
| Sandoval-Calderón M, Nechitaylo T, Engl T, Vogel H, Koehler S, Kaltenpoth M | 2019 | RNA-Seq of Philanthus triangulum: female antenna rRNA depleted, polyA-enriched | https://www.ncbi.nlm.nih.gov/biosample/11793601 | Genbank, SAMN11793601 |

The following previously published datasets were used:

| Author(s) | Year | Dataset title | Dataset URL | Database and Identifier |
|---|---|---|---|---|
| Regulski M, Tully T | 1995 | Drosophila melanogaster Ca/calmodulin-dependent nitric oxide synthase (NOS) mRNA, complete cds | https://www.ncbi.nlm.nih.gov/nuccore/U25117.1 | GenBank, U25117.1 |
| Watanabe T, Shiga T, Yamamoto T, Suzuki N, Ito E | 2005 | Apis mellifera AmNOS mRNA for nitric oxide synthase, complete cds | https://www.ncbi.nlm.nih.gov/nuccore/AB204558.1 | GenBank, AB204558.1 |
| Luckhart S, Vodovotz Y, Cui L | 1999 | Anopheles stephensi nitric oxide synthase gene, complete cds | https://www.ncbi.nlm.nih.gov/nuccore/AH007775.1 | GenBank, AH007775.1 |
| Yuda M | 1996 | Rhodnius prolixus salivary gland nitric oxide synthase mRNA, complete cds | https://www.ncbi.nlm.nih.gov/nuccore/U59389.1 | GenBank, U59389.1 |
| Nighorn A, Gibson NJ, Rivers DM, Hildebrand JG, Morton DB | 1998 | Manduca sexta nitric oxide synthase (NOS) mRNA, complete cds | https://www.ncbi.nlm.nih.gov/nuccore/AF062749.1 | GenBank, AF062749.1 |
| Werren et al | 2016 | Nasonia vitripennis nitric oxide synthase (Nos), mRNA | https://www.ncbi.nlm.nih.gov/nuccore/NM_001168232.1 | GenBank, NM_001168232.1 |

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

# Appendix 1

DOI: https://doi.org/10.7554/eLife.43718.029

## Additional data 1: Bioassay for an antifungal effect of paralyzed honeybees

To test whether paralyzed bees alone show some antifungal activity, we conducted an additional experiment. Petri dishes (N = 12) inoculated with conidia of Aspergillus flavus isolated from brood cells were prepared as described in the main text. Bees from brood cells with two bees were separately placed on the agar. Each of the bees as well as a control area without bee was covered by a cap with a volume of 3 ml (similar to a typical brood cell). Petri dishes were incubated at 25°C in a dark climate chamber. After 24 h fungus growth on the area covered by each of the caps was categorized as: 0 = no fungus, 1 = very little fungus (only few hyphae at the rim), 2 = little fungus (several hyphae at the rim), 3 = strong fungus (area overgrown with hyphae) (*Appendix 1—table 1*). We tested for overall differences in fungus growth among the three treatment groups using a Friedman test (Software SPSS Version 24). Subsequently, pairwise comparisons were conducted using Bonferroni-corrected Wilcoxon tests.

**Appendix 1—table 1.** Scores of fungus growth (0 = no fungus, 1 = very little fungus, 2 = little fungus, 3 = strong fungus) after 24 h of incubation at 25°C on Petri dishes inoculated with Aspergillus flavus conidia. Scores are given for three areas of the petri dish that were covered with a cap under which either a 'Bee with egg', a 'Bee without egg' was placed as well as a 'Control' with no bee.

| Petri dish | Bee with egg | Bee without egg | Control |
|---|---|---|---|
| 1 | 0 | 3 | 3 |
| 2 | 0 | 2 | 2 |
| 3 | 0 | 2 | 3 |
| 4 | 1 | 2 | 3 |
| 5 | 1 | 3 | 3 |
| 6 | 0 | 3 | 3 |
| 7 | 0 | 3 | 3 |
| 8 | 0 | 3 | 3 |
| 9 | 0 | 3 | 3 |
| 10 | 0 | 3 | 3 |
| 11 | 0 | 3 | 3 |
| 12 | 0 | 3 | 3 |

DOI: https://doi.org/10.7554/eLife.43718.030

The medians (and quartiles) were 0 (0, 0) for the 'bee with egg', 3 (2.25, 3) for the 'bee without egg' and 3 (3, 3) for the control area without bee. There was a significant overall difference among medians (Friedman test: chi2(tie corrected) = 22.9, D.F. = 2, P = 0.000019). Pairwise comparisons revealed significant differences between the group "bee with egg" and both others (p = 0.0022 in both cases). The groups 'bee without egg' and 'control' did not differ significantly (p = 1.0). The results show that under the cap with the bee carrying an egg, fungus growth is nearly completely inhibited, whereas the areas with a bee not carrying an egg and the controls did not differ. Thus, there is no evidence that a paralyzed bee produces any antifungal gas or relevant amounts of NO⋅.

**Appendix 1—table 2.** Hazard ratios (and 95 % confidence intervals) for the comparison of timing of the onset of fungus growth on bees of four treatment groups: bees that carried an egg and were embalmed (+/+), bees that carried no egg but were embalmed (-/+), bees that carried no egg and were not embalmed (-/-) and bees that carried an egg but were not embalmed (+/-).

**Comparison**

| Egg/Embalming | Egg/Embalming | Hazard ratio | 95% conf. interval |
|---|---|---|---|
| +/+ | -/+ | 0.07 | 0.19 - 0.03 |
| +/+ | -/- | 0.13 | 0.36 - 0.05 |
| +/+ | +/- | 0.65 | 0.90 - 0.47 |
| -/+ | -/- | 0.48 | 0.79 - 0.29 |
| +/- | -/+ | 0.61 | 0.45 - 0.83 |
| +/- | -/- | 0.22 | 0.10 - 0.47 |

DOI: https://doi.org/10.7554/eLife.43718.031

# Additional discussion 1

Oxygen content of air is 21% (=210000ppm). Thus, in a brood cell of 3.2 ml there are about 670µl oxygen. Based on (limited) available data on oxygen uptake of Hymenoptera eggs (0.8µl O2/mg/h for a hymenopteran parasitoid egg (*Fisher, 1963*), a beewolf egg of 5mg would need 4µl O2/h. Living honeybees take up about 0.2µl O2/mg/h (*Allen, 1959*), thus a honeybee worker of about 80mg will consume 16µl O2/h. Consequently, the oxygen available in a brood cell should be sufficient to support both, metabolism of the egg and oxidation of NO·. Moreover there will be an influx of oxygen form the surrounding sand by diffusion along an emerging gradient. Thus, NO· is certainly oxidized to $NO_2$·. in beewolf brood cells. The rate of this conversion can, however, not be specified at the moment.

