## [Decision Letter]

Thank you for submitting your article "Nitric oxide radicals are emitted by wasp eggs to kill mold fungi" for consideration by *eLife*. Your article has been reviewed by three peer reviewers, one of whom is a member of our Board of Reviewing Editors, and the evaluation has been overseen by a Reviewing Editor and Ian Baldwin as the Senior Editor. The reviewers have opted to remain anonymous.

The reviewers have discussed the reviews with one another and the Reviewing Editor has drafted this decision to help you prepare a revised submission.

Your manuscript addresses a very interesting study on how a beewolf preserves the bees that it has collected and protects them from degenerating by the provisioning of NO· by the eggs. All three reviewers provide positive evaluations of your study. Yet, some aspects require further improvement, including an essential additional experiment.

The main issues to be resolved are:

1) The study focuses on NO·, thereby potentially missing other active compounds as highlighted by reviewer 2. To assess the role of NO·, additional evidence to support NO· as being the causal agent is needed, i.e. eliminating NO· with the inhibitor as suggested by reviewer 3. This additional experiment will provide essential information to conclude on the effect of NO· within the total characteristics of the egg.

2) The discussion on the NO· levels in the brood cell, leaking in the soil, survival of larvae and later developmental stages of the wasp, survival of the bees beyond day 3 etc. is not satisfactory. See comments by reviewers 1 and 2. Especially the brief mentioning of mortality of larvae needs more attention and clarification as this is vital for understanding the functioning of NO· as a true resource protectant.

3) The discussion on the NOS gene and its uniqueness in the beewolf. Please clarify what makes the detected NOS gene substantially different from its orthologues and therefore capable of performing a different function (higher NOS production), and better address the option that NO· might arise from different genes.

In addition, the reviewers have made other valuable comments that will be helpful in further improving the manuscript.

Reviewer #1:

This manuscript provides a very interesting study of how beewolf eggs may prevent bee supplies from fungal colonisation by producing high levels of NO· radicals. The study shows that the presence of beewolf eggs prevents mold growth on paralyzed honeybees in the brood cells, that NO· is produced in the eggs, is present around the paralyzed bees and that eggs prevent the growth of a fungus, i.e. *Aspergillus flavus*. Yet, some important questions remain to fully understand how NO· radicals protect the bees while not harming the beewolf larva or its symbiotic bacteria.

Although the evidence supports the beewolf egg as a source of NO·, the data cannot exclude that NO· is also produced by the paralyzed bee on which the egg has been deposited. The data presented in Figure 2B would support this, as well as the data in Figure 4B. Why was no experiment included in which mold growth on a bee was recorded in the presence of an egg that was presented without a bee substrate to supplement the data in Figure 2B? The data in Figure 4B show that NO· is present on a bee on which an egg has been laid but it cannot be excluded that part of this is produced by the bee.

The manuscript does not present all data and some crucial data are presented as supplementary figures. Why are not all crucial data presented in the main text? For instance (a) the data for the 5 additional fungi are not presented at all (Results, first paragraph), (b) Subsection “Identification of the antifungal volatile”, last paragraph, please show these data, there is no space limitation, (c) Subsection “Identification of the antifungal volatile”, last paragraph: why is the data of the Griese analysis of the headspace samples not presented?, (d) Figures 9 and 10 provide support for the production of NO· in the beewolf egg; the picture (Figure 9) is a combination of two pictures (why?); what is the age of the egg shown in Figure 9 (very important in the context of Figure 10 and Figure 5)? (e) fungus growth inhibition was recorded at 24, 48 and 72 hours (subsection “Fungus inhibition assays”, last paragraph) but no data for 48 or 72 h are presented.

In Figure 6, please provide the exact P value for the comparison between control and D-NAME instead of stating that it is > 0.05.

The picture in Figure 3 presents only one replication. How many replicates were there and how similar were the results? Preferably the pictures of the replicates should be presented in a supplementary figure. The number of replications is low for some experiments, e.g. for timing of NO· emission only 4 replicates were used but for the iodometry experiment a large number of eggs (233) has been used.

Please add a note on the number of replications to the legend of Figure 6 and Figure 10.

The discussion on how the beewolf eggs and the *Streptomyces* bacteria survive the high NO· levels is not convincing to me. The authors first argue that the NO· concentrations in the brood cell are way too high for fungi to develop resistance (Discussion, twelfth paragraph) but then state that potentially the *Streptomyces* bacteria are resistant to the NO·, maybe helped by being at the ceiling of the brood cell. However, given the dimensions and the average very high NO· concentration the concentration near the ceiling is likely still very high. Moreover, the arguments given for how the beewolf larvae may survive the high concentrations (Discussion) are not convincing; after all they are the source of the high levels being emitted and even as larva they are killed by these high concentrations.

How do the larvae that emerge from the NO· producing eggs survive in the brood cell? The comment in the twelfth paragraph of the Discussion suggests they do not, which would imply that the NO· concentration drops with time if the larvae are to survive. If that is true (is it?) then how are the bees protected from molding during days 3-11 when the larva develops (Figure 1)? The study provides information on NO· production with time but not on NO· concentration during the 11 days of larval development, which would be needed for this.

The manuscript should provide answers to these questions to be fully convincing on the protective effect of egg-produced high NO· concentrations in terms of preventing their food to be infested by molds while they themselves should survive the intoxicated brood cell.

Reviewer #2:

In this interesting manuscript, Strohm et al. propose a new mechanism of brood defense from fungal pathogens based on the production of antifungal volatile compounds. This is a novel and potentially important idea that offers new insights into how insects might combat pathogens during vulnerable life cycle stages. On the whole, I enjoyed reading this well-written manuscript and the ideas that it contains. However, I was left with some questions that I hope the authors can address:

1) Given the existence of other defense mechanisms in the beewolf system, is there any potential for NO production to interact synergistically or antagonistically with these other defense mechanisms? For example, are there any compounds on the eggs themselves that might provide some antifungal activity? A broader discussion (e.g., in the Introduction) of the various beewolf defense mechanisms and their importance in different phases of the beewolf life cycle might help clarify these issues. This issue also intersects with the discussion of NO resistance by the *Streptomyces* symbiont, such that there might be trade-offs between defenses active at different life cycle stages.

2) The author's focus on NO is quite directed, and so might cause them to miss other potential volatile compounds that are also involved in the observed bioactivities. Better justifying this approach would be valuable. There also seems to be some disconnect between the first paragraph of the subsection “Identification of the antifungal volatile”, where the authors seem to interpret the odor given off by the beewolf eggs as similar to NO_2_, and the first paragraph of the subsection “Identification of the antimicrobial volatile”, where samples where nitrate was reduced to detect NO_2_ did not differ from those that were not reduced, implying that only NO (and not NO_2_) was produced by the eggs, despite the odor.

3) The authors' calculations of likely NO concentrations in brood cells is very interesting, but I am unsure how likely these values are to relate to the actual values in the wild. In particular, how likely is it that there is "no loss due to reactions or leaking out of the confined space of brood cells”? It seems that soil could conceivably be quite porous and so allow NO to escape the brood cell. A pulse that diffuses quickly would impact the need for and nature of NO resistance mechanisms that the authors discuss extensively. Similarly, what is the likely soil temperature of a typical beewolf brood cell? The assay temperatures strike me as being warmer than many soils in temperate Europe, although I imagine that the insect could select appropriate sites to meet this constraint. Finally, the authors propose oxic mechanisms for NO (bio)chemistry (e.g., conversion of NO to NO_2_) without specifying if the brood cells are oxic or anoxic. Reduced oxygen in brood cells seems plausible if they are sealed while the egg is developing (and presumably respiring).

4) The authors give several potential mechanisms for NO resistance, but it is unclear how plausible these are given the potentially high concentrations of NO that might be present. A more quantitative discussion of NO resistance would be helpful here, e.g., are the proposed resistance mechanisms likely to be sufficient under the proposed levels of NO accumulation?

5) Discussion, twelfth paragraph: the authors propose that coating the symbiotic *Streptomyces* in hydrocarbons might protect them from NO toxicity. However, the bees are also coated in hydrocarbons (Introduction, fourth paragraph). Wouldn't the resistance mechanism proposed for the bacteria then also protect fungi growing on the bees? Antioxidants are also proposed to be mixed with these hydrocarbons – how likely are these to exist?

6) Discussion, fifteenth paragraph: the observation that larvae are not resistant to NO produced by the eggs is concerning. Are other life cycle stages similarly sensitive to NO, e.g., eggs exposed before they start producing NO or immediately after? The sensitivity of larvae might imply that NO does not accumulate in the brood cells for very long (see also my comment #3 above).

7) I am not entirely clear how to interpret Figure 9 because I am unclear what baseline level of staining to expect before NOS induction. Comparisons between time points and/or other species where NOS is not thought to be induced would provide useful controls for this experiment.

8) The focus on the detected NOS gene as being responsible for the observed NOS activity makes the argument that this gene behaves differently compared to all other known genes without direct evidence for such divergent behavior. The one inferred splice variant by itself seems to provide weak evidence for higher activity, given that higher NOS activity observed in humans is due to different genes (vs. splice variants of the same gene) and that NOS activity in humans is of a different order of magnitude than implied here for beewolves. The inhibition experiment provides some evidence for NOS being involved, although there is still the assumption that NOS is the only enzyme affected by the inhibitor. Differential gene expression matching NO production, direct assays of the NOS variants, or even signatures of adaptive evolution (e.g., Dn/Ds metrics) would provide more convincing support for the uniqueness of the beewolf NOS, especially when combined with explicit comparisons to related species that due not produce similarly high levels of NO as controls.

*Reviewer #3:*

This is a very nice paper that touches on different fascinating aspects of insect's biology, from an ecological to a molecular point of view, and should interest a broad audience.

This study shows that eggs of the parasitic wasp Philanthus triangulum emit high amounts of nitric oxide as antifungal compounds to protect themselves and preserve the food source of their progeny. The enzyme responsible for NO production is identified (NOS) and a differential splicing event between egg and adult forms may explain the high levels observed in eggs. These are novel and quite interesting findings that complement other defense strategies that have been previously reported in insects. The story is well written and nicely presented. Data are convincing. The Discussion is complete and carefully addresses the novelty and the biochemical and biological implications of the findings.

That eggs emit a volatile compound that has antifungal activity is demonstrated by 1) showing that host bees not in direct or indirect contact with *P. triangulum* eggs get more infected with mold fungi, 2) showing that eggs release an antifungal volatile when placed above a culture of different fungal strains growing on agar plates. That eggs emit NO is demonstrated by histochemical and colorimetric methods and by using fluorescent probes. However, that egg-released NO is responsible for inhibiting fungal growth is only supported by the correlation between the above observations and by previous reports on the mycotic activity of NO. I see the difficulty to genetically prove the causality of NO production in the antifungal effect of egg-derived volatile production. A simple additional experiment may however help to strengthen this hypothesis. It would be important to inhibit NO release in eggs by using L-NAME, the NOS-inhibiting substance used for Figure 6, and to show that egg-induced inhibition of fungal growth on agar plate (as in Figure 3) is abolished or greatly diminished.

[Editors' note: further revisions were requested prior to acceptance, as described below.]

Thank you for submitting your article "Nitric oxide radicals are emitted by wasp eggs to kill mold fungi" for consideration by *eLife*. Your article has been reviewed by a Reviewing Editor and Ian Baldwin as the Senior Editor.

In response to the reviews you have carried out an additional experiment and have modified the manuscript in various places. As a result the manuscript has been clearly improved. Yet, a few issues still remain.

The additional experiment tests the effect of synthetic NO on fungal infection of honeybees and shows that synthetic NO protects the bees from infection. This experiment was done for logistic reasons as an alternative to the suggested experiment (eliminating NO· from the natural mixture with the inhibitor). The result underlines the involvement of NO but leaves the option of other compounds being involved open. One of the reviewers had commented that the focus on NO is quite directed, and so might result in missing other potential volatile compounds that are also involved in the observed bioactivities. In response to the question to better justify the focus on NO you comment that is not reasonable. If the suggested experiment had been done that might be the case but because the involvement of other compounds has not been excluded I consider it reasonable to explain why you focussed on NO.

In response to the comment that "the observation that larvae are not resistant to NO produced by the eggs is concerning." you respond with comments in the response letter but this has not resulted in a change in the manuscript. It will be important to provide information in the manuscript itself why the larvae will not suffer from NO as this is vital to the understanding of the protective activity of the NO in conserving a resource. This aspect was also identified by one of the other reviewers.

---

## [Author Response]

Your manuscript addresses a very interesting study on how a beewolf preserves the bees that it has collected and protects them from degenerating by the provisioning of NO· by the eggs. All three reviewers provide positive evaluations of your study. Yet, some aspects require further improvement, including an essential additional experiment.The main issues to be resolved are:1) The study focuses on NO·, thereby potentially missing other active compounds as highlighted by reviewer 2. To assess the role of NO·, additional evidence to support NO· as being the causal agent is needed, i.e. eliminating NO· with the inhibitor as suggested by reviewer 3. This additional experiment will provide essential information to conclude on the effect of NO· within the total characteristics of the egg.

We have conducted an additional experiment to assess whether NO can be responsible for the strong antifungal effect of the gas emitted by beewolf eggs. Since we do not have beewolves available during winter and spring due to the obligatory diapause of this insect species, we were not able to do the inhibitory experiment as suggested by reviewer 3. Instead, we applied synthetic NO to artificial brood cells with bees in amounts that mimicked the estimated concentrations in natural brood cells. There was a significant delay of fungus growth on the bees that were subjected to the NO compared to respective controls. Moreover, comparison of the hazard ratio (the appropriate measure of effect size for such survival analyses) calculated for this experiment and a similar experiment that was previously conducted with beewolf eggs instead of synthetic NO revealed similar values with mutually overlapping 95% confidence intervals. (side note: It is not meaningful to compare the data of the two experiments directly, since they were conducted at different times with different bees and possibly different strains of fungi; therefore we compared the effect sizes obtained in the two experiments). Thus, the experiment shows equivalent antifungal effects of artificial and beewolf-produced nitric oxide in situ, and there was no evidence for an additional agent being active in the gas emitted by beewolf eggs.

2) The discussion on the NO· levels in the brood cell, leaking in the soil, survival of larvae and later developmental stages of the wasp, survival of the bees beyond day 3 etc. is not satisfactory. See comments by reviewers 1 and 2. Especially the brief mentioning of mortality of larvae needs more attention and clarification as this is vital for understanding the functioning of NO· as a true resource protectant.

Please see our comments below.

3) The discussion on the NOS gene and its uniqueness in the beewolf. Please clarify what makes the detected NOS gene substantially different from its orthologues and therefore capable of performing a different function (higher NOS production), and better address the option that NO· might arise from different genes.

Please see our comments below.

In addition, the reviewers have made other valuable comments that will be helpful in further improving the manuscript.Reviewer #1:This manuscript provides a very interesting study of how beewolf eggs may prevent bee supplies from fungal colonisation by producing high levels of NO· radicals. The study shows that the presence of beewolf eggs prevents mold growth on paralyzed honeybees in the brood cells, that NO· is produced in the eggs, is present around the paralyzed bees and that eggs prevent the growth of a fungus, i.e. Aspergillus flavus. Yet, some important questions remain to fully understand how NO· radicals protect the bees while not harming the beewolf larva or its symbiotic bacteria.Although the evidence supports the beewolf egg as a source of NO·, the data cannot exclude that NO· is also produced by the paralyzed bee on which the egg has been deposited. The data presented in Figure 2B would support this, as well as the data in Figure 4B. Why was no experiment included in which mold growth on a bee was recorded in the presence of an egg that was presented without a bee substrate to supplement the data in Figure 2B? The data in Figure 4B show that NO· is present on a bee on which an egg has been laid but it cannot be excluded that part of this is produced by the bee.

We indeed cannot completely rule out the possibility that paralyzed honeybees emit small amounts of nitric oxide. However, we added the results of an experiment that shows that a paralyzed honeybee does not delay fungus growth and thus does not produce relevant amounts of NO. We present this experiment in the SI section (SI Additional data 1) in order not to overload the main text. Thus, our results show that (i) beewolf eggs produce incredibly high concentrations of NO, while (ii) NO emission from paralyzed bees (if it indeed occurs at all, which seems very unlikely given that NO emission has never been shown previously for an adult insect) are ecologically negligible.

The manuscript does not present all data and some crucial data are presented as supplementary figures. Why are not all crucial data presented in the main text? For instance a) the data for the 5 additional fungi are not presented at all (Results, first paragraph).

It is stated in the text that for the bioassays with 5 fungi (N=8 replicates each), in all cases there was a clear difference with no fungus growing in the area exposed to the gas emanating from the egg and fungus growth in the control area (as exemplified in Figure 3). Unfortunately, we have not taken photos of all these experiments, since the results were so clear-cut. We have changed the text to clarify this. Concerning other figures, we opted not to overload the main text with figures, so we left additional data supporting – but not directly critical for – the main points of the study in the supplementary data.

b) Subsection “Identification of the antifungal volatile”, last paragraph, please show these data, there is no space limitation.

Since pictures of the eggs are presented, only a picture of an injected larva is not shown. We initially did not add these, because they are somewhat blurred due to the larvae moving during exposure. Anyhow, we have now added pictures of a larva to the SI file. Moreover, we provide all pictures of all eggs and larvae injected with the fluorescent probe as a source data file.

c) Subsection “Identification of the antifungal volatile”, last paragraph: why is the data of the Griese analysis of the headspace samples not presented?

We are not sure what the reviewer means here. The Griess assay as employed here was to prove the existence of nitrite that can, with the setup that we applied only be generated by a gas, either NO or NO_2_, emitted from the egg. We have now added an original picture of vials showing the positive reaction. We did not use the Griess reaction for quantification, since this assay is not suitable for prolonged sampling as would be necessary for the analysis of the emission from beewolf eggs. The picture (Figure 4—figure supplement 1) is not very appealing since it was meant for documentation, not for publication, but it clearly shows the color change in the Griess reagent exposed to the headspace of beewolf eggs.

d) Figures 9 and 10 provide support for the production of NO· in the beewolf egg. The picture (Figure 9) is a combination of two pictures (why?).

The egg was too large to fit into the field of view of the camera and we did not have a lower magnification lens available, so we had to take two pictures (basically the left and the right part) and stitched these. This is stated more clearly now.

What is the age of the egg shown in Figure 9 (very important in the context of Figure 10 and Figure 5)?

The egg was fixed 15-16h after oviposition, the time of peak NO production. This is stated now in the revised manuscript.

e) fungus growth inhibition was recorded at 24, 48 and 72 hours (subsection “Fungus inhibition assays”, last paragraph) but no data for 48 or 72 h are presented.

We suspect that the line numbers the reviewer refers to differ from the PDF that we provided (which is the same as the one that can be downloaded from the *eLife* site). So we assume that the reviewer refers to the last paragraph of the subsection “Fungus inhibition assays”, It is stated that the results for 48 and 72 h are the same as for 24 h. We have changed the text to clarify this.

In Figure 6, please provide the exact P value for the comparison between control and D-NAME instead of stating that it is > 0.05.

The P value was added to the figure.

The picture in Figure 3 presents only one replication. How many replicates were there and how similar were the results? Preferably the pictures of the replicates should be presented in a supplementary figure.

The number of replicates for the *Aspergillus* isolates from brood cells (N = 20) and for the five additional fungi (N=8) were given in the text. Unfortunately, as mentioned above, we did not take pictures of all the petri dishes of these experiments. The results were clear-cut, with no exceptions and no intermediate cases. This is also stated in the Materials and methods, but is now better explained in the revised text.

The number of replications is low for some experiments, e.g. for timing of NO· emission only 4 replicates were used but for the iodometry experiment a large number of eggs (233) has been used.

As stated in the paragraph "General experimental procedures" in the Materials and methods section, availability of beewolf eggs is often limited and difficult to predict. It is particularly challenging to obtain large sample sizes, if the time of oviposition has to be known and the egg has to be collected shortly thereafter, as is the case for the experiment to reveal the timing of NO emission. Beewolves lay their eggs mostly during the night, thus monitoring oviposition is rather laborious. We managed to measure 4 replicates for the whole period (0-22h), these were very similar. We did several additional measurements of other eggs that did, however, not span the whole period. We have now included all these measurements and modified the figure accordingly.

Knowing that NO production within the first hours after oviposition is rather low, it was comparatively much easier to produce a large sample for the quantification of NO emission.

Please add a note on the number of replications to the legend of Figure 6 and Figure 10.

We added the sample sizes to the legend of Figure 6.

The discussion on how the beewolf eggs and the Streptomyces bacteria survive the high NO· levels is not convincing to me. The authors first argue that the NO· concentrations in the brood cell are way too high for fungi to develop resistance (Discussion, twelfth paragraph) but then state that potentially the Streptomyces bacteria are resistant to the NO·, maybe helped by being at the ceiling of the brood cell. However, given the dimensions and the average very high NO· concentration the concentration near the ceiling is likely still very high.

We agree that the survival of both, the eggs and the bacteria are surprising taking into account the high concentrations of nitrogen oxides. At the moment we can only speculate about the ultimate causes and the proximate mechanisms.

With regard to the eggs: As has been stated in the manuscript, there might be no "free" NO in the egg tissue. Possibly, the NO is bound to carrier molecules that take them to the egg shell for release. After oxidation to NO_2_, diffusibility is considerably reduced compared to NO so that NO_2_ might not be able to penetrate the egg shell. The larva however, has only a very thin cuticle and gas exchange happens by means of tracheae. Thus NO_2_ might enter easily and kill the larva. In addition, the eggs likely mount countermeasures to protect itself against NO (e.g. controlling reactive oxygen species that in situ drastically enhance the detrimental effect of NO), while the larvae won’t do this under natural conditions.

With regard to the symbionts, one important ultimate aspect that was not made clear enough in the manuscript is that the symbiotic bacteria coevolved with beewolves; meaning the same bacterial lineages were repeatedly exposed to NO and were thus subject to strong selection. The fungi on the other hand that are to be found in beewolf brood cells are unspecialized ubiquitous and opportunistic. Thus there is no repeated exposition of the same fungal strains to high concentrations of NO. Consequently, there is negligible selection for resistance.

There might be several coactive mechanisms that protect the bacteria: elevated location in the brood cell, screening by the lipids of the surrounding matrix, antioxidants in the surrounding matrix, molecular detoxification of NO and/or NO_2_. These aspects have been discussed more explicitly and carefully in the revised manuscript, and future experiments will be targeted towards elucidating the resistance of both beewolf and symbionts against NO.

Moreover, the arguments given for how the beewolf larvae may survive the high concentrations (Discussion) are not convincing; after all they are the source of the high levels being emitted and even as larva they are killed by these high concentrations.

We suspect that the reviewer does not refer to larvae but to eggs as producers of NO here, so please see our response to the preceding comment.

How do the larvae that emerge from the NO· producing eggs survive in the brood cell? The comment in the twelfth paragraph of the Discussion suggests they do not, which would imply that the NO· concentration drops with time if the larvae are to survive.

Since the brood cell is constructed in sandy soil, there is certainly a gradual loss of nitrogen oxides due to diffusion out of the brood cell and dissolution in water. However, this leakage of nitrogen oxides is most probably slow, so that during peak NO production the concentrations will be high. Actually, the typical smell of NO_2_ disappears by the time the larvae hatch, so larvae are not exposed to the nitrogen oxides (at least not to such high concentrations).

If that is true (is it?) then how are the bees protected from molding during days 3-11 when the larva develops (Figure 1)?

As is stated in the manuscript, the brood cell fumigation considerably reduces the detrimental effect of fast growing mold fungi by killing them. Thus the major threat to the larvae is eliminated. Spores that germinate after the nitrogen oxide levels have dropped or are not exposed to the gas because they are covered by the bee's body might be able to grow and compete with the larvae. As is shown by the added experiment, embalming of the bees corroborates the antifungal effect of fumigation. However, even the combined protective effect of the fumigation and the embalming is, of course, not perfect. In fact, we have already reported that cocoons from brood cells that show some fungus infestation are smaller than cocoons from not infested brood cells (Strohm, 2000). However, in most cases, larvae will be able to complete feeding and spin their cocoons. Then the antibiotics on the cocoon walls that are produced by the symbiotic *Streptomyces* bacteria will take over the protection of the larvae.

The study provides information on NO· production with time but not on NO· concentration during the 11 days of larval development, which would be needed for this.

It is very difficult to measure NO over time on such a small spatial scale in situ. Although the concentrations in the brood cell are high, the absolute amounts are small. There are devices to measure NO in tissue but, according to the manufacturers these only work well in aqueous surroundings. So, yes, unfortunately, we do not have data on the NO concentration in the brood cell over time. As stated above, however, the smell of NO_2_ disappears before larvae hatch. Hence, it appears that the short burst of NO production serves to kill fungi, and the NO then diffuses slowly out of the brood cell, leaving a sanitized environment that allows the larva to hatch successfully.

The manuscript should provide answers to these questions to be fully convincing on the protective effect of egg-produced high NO· concentrations in terms of preventing their food to be infested by molds while they themselves should survive the intoxicated brood cell.

We believe that the manuscript provides a thorough description of a novel antimicrobial strategy in a ground-developing insect, showing that the production of high concentrations of NO reduces fungal infestation of the beewolf’s provisions in vitro and in situ. We have added additional data supporting the sanitizing role of NO in situ, and expanded on the discussion of how the beewolf egg and the symbiotic bacteria survive NO exposure in the revised manuscript.

Reviewer #2:In this interesting manuscript, Strohm et al. propose a new mechanism of brood defense from fungal pathogens based on the production of antifungal volatile compounds. This is a novel and potentially important idea that offers new insights into how insects might combat pathogens during vulnerable life cycle stages. On the whole, I enjoyed reading this well-written manuscript and the ideas that it contains. However, I was left with some questions that I hope the authors can address:1) Given the existence of other defense mechanisms in the beewolf system, is there any potential for NO production to interact synergistically or antagonistically with these other defense mechanisms? For example, are there any compounds on the eggs themselves that might provide some antifungal activity? A broader discussion (e.g., in the Introduction) of the various beewolf defense mechanisms and their importance in different phases of the beewolf life cycle might help clarify these issues. This issue also intersects with the discussion of NO resistance by the Streptomyces symbiont, such that there might be trade-offs between defenses active at different life cycle stages.

We already had data available on the possible interaction of the fumigation with the embalming of the honeybees with hydrocarbons. We had not presented these data in this manuscript because we believed it to be beyond the scope of this study. However, because of the reviewer's comment, we added these data to the revised manuscript. In fact, embalming and fumigation together have a stronger antifungal effect than either mechanism alone. The interaction with the symbiotic bacteria is of course extremely interesting and is currently under investigation. We added a brief description of the antimicrobial symbiosis of beewolves to the Introduction.

2) The author's focus on NO is quite directed, and so might cause them to miss other potential volatile compounds that are also involved in the observed bioactivities. Better justifying this approach would be valuable.

There are some other strong gaseous oxidants. However, as a result of many preliminary approaches and considerations, NO was, by far, the most plausible alternative. We do think that a detailed justification of the focus on NO is not reasonable.

There also seems to be some disconnect between the first paragraph of the subsection “Identification of the antifungal volatile”, where the authors seem to interpret the odor given off by the beewolf eggs as similar to NO_2_, and the first paragraph of the subsection “Identification of the antimicrobial volatile”, where samples where nitrate was reduced to detect NO_2_ did not differ from those that were not reduced, implying that only NO (and not NO_2_) was produced by the eggs, despite the odor.

The odor definitely is caused by NO_2_; as stated, NO is odorless (though being highly reactive, it has less oxidative power than NO_2_). The Griess assay resulted in red coloration both with and without reduction. The statement of no difference between the groups meant that both showed the red coloration. Because the red azo dye that is formed is not perfectly stable over time, as stated by the manufacturer's instructions of the kit, quantitative measurements for extended exposure times are not recommended. This is now stated in the revised version.

3) The authors' calculations of likely NO concentrations in brood cells is very interesting, but I am unsure how likely these values are to relate to the actual values in the wild. In particular, how likely is it that there is "no loss due to reactions or leaking out of the confined space of brood cells”? It seems that soil could conceivably be quite porous and so allow NO to escape the brood cell. A pulse that diffuses quickly would impact the need for and nature of NO resistance mechanisms that the authors discuss extensively.

As stated above, we unfortunately have no actual data on the time course of NO/NO_2_ concentrations in the brood cells. We discuss the issue of leakage in more detail now. Beewolves construct their brood cells in rather compact sandy soil. Moreover, the walls of the nest burrows and the brood cells are covered with a layer of hydrocarbons that might provide an additional barrier; this is now mentioned in the revised text. So brood cells are neither perfectly sealed nor are they porous enough to let gases leak out easily. Because there is no airflow within the brood cell, removal of NO/NO_2_ will happen by diffusion only. Since most NO is produced within a short 2h burst, it will accumulate to high concentrations that are detrimental to microbes. The time course of diffusion into the surrounding soil remains thus far unknown und hard to quantify.

Similarly, what is the likely soil temperature of a typical beewolf brood cell? The assay temperatures strike me as being warmer than many soils in temperate Europe, although I imagine that the insect could select appropriate sites to meet this constraint.

Of course temperatures in the field vary somewhat due to differences in depth of the brood cells, sand structure, sand moisture, inclination and orientation of the nest site. Beewolves establish their nests only at very sunny sites and sandy soil and the incubation temperatures used in the experiments reflect a representative temperature of natural brood cells {Herzner and Strohm, 2008). Moreover, larval development is best around 25-30°C (Figure 5 in Strohm, 2000). This information has now been added to the revised version.

Finally, the authors propose oxic mechanisms for NO (bio)chemistry (e.g., conversion of NO to NO_2_) without specifying if the brood cells are oxic or anoxic. Reduced oxygen in brood cells seems plausible if they are sealed while the egg is developing (and presumably respiring).

This issue is discussed in more detail now in the SI file.

4) The authors give several potential mechanisms for NO resistance, but it is unclear how plausible these are given the potentially high concentrations of NO that might be present. A more quantitative discussion of NO resistance would be helpful here, e.g., are the proposed resistance mechanisms likely to be sufficient under the proposed levels of NO accumulation?

We are not sure whether the reviewer refers to a possible resistance of the symbiotic bacteria or the beewolf egg itself.

With regard to the bacteria we changed the text to make clear that at the moment we can only speculate about possible mechanisms and added some references. Since a more quantitative analysis is experimentally also elaborate, the detailed reaction of the symbiotic bacteria will be the focus of a future study.

With regard to beewolf eggs, we mention nitrophorins that might carry the NO to the egg shell, thus little "free" NO would occur in the tissue of the embryo. As stated in the text, such nitrophorins are known from blood sucking bugs. These nitrophorins might be produced in amounts sufficient for the transport of NO. However, we do not find it meaningful to further speculate about this and other mechanisms. Investigation of these mechanisms goes far beyond the scope of the present manuscript.

5) Discussion, twelfth paragraph: the authors propose that coating the symbiotic Streptomyces in hydrocarbons might protect them from NO toxicity. However, the bees are also coated in hydrocarbons (Introduction, fourth paragraph). Wouldn't the resistance mechanism proposed for the bacteria then also protect fungi growing on the bees? Antioxidants are also proposed to be mixed with these hydrocarbons – how likely are these to exist?

Spores embedded in the CHC layer on the bees might indeed be protected against NO. However, these would also be shielded from liquid water that would be necessary for germination, so these would not be dangerous for beewolf progeny. Spores that are not fully covered by hydrocarbons might germinate but will likely be affected by the NO.

With regard to the antioxidants, we have preliminary data that antioxidants are present in the matrix surrounding the symbionts in the brood cell. However, this is again beyond the scope of the current manuscript and needs substantial additional experimental validation.

6) Discussion, fifteenth paragraph: the observation that larvae are not resistant to NO produced by the eggs is concerning. Are other life cycle stages similarly sensitive to NO, e.g., eggs exposed before they start producing NO or immediately after? The sensitivity of larvae might imply that NO does not accumulate in the brood cells for very long (see also my comment #3 above).

As stated above nitrogen oxides will slowly disappear from the brood cells, thus, larvae are not harmed under natural conditions. The mentioned cases of larvae dying from exposition to NO/NO_2_ accidentally occurred during early experiments to identify the gas.

We have not tested whether early egg stages are sensitive against NO/NO_2_.

7) I am not entirely clear how to interpret Figure 9 because I am unclear what baseline level of staining to expect before NOS induction. Comparisons between time points and/or other species where NOS is not thought to be induced would provide useful controls for this experiment.

Since the method is rather complicated we analyzed only few eggs in this way. According to the literature, without fixation, several enzymes show NADPH diaphorase activity. However, with fixation, blue staining indicates sites of NOS activity. Since the embryonic tissue of this egg (about 15h after oviposition, i.e. during the time of peak NO production) is clearly stained blue, this provides clear evidence for NOS activity.

8) The focus on the detected NOS gene as being responsible for the observed NOS activity makes the argument that this gene behaves differently compared to all other known genes without direct evidence for such divergent behavior. The one inferred splice variant by itself seems to provide weak evidence for higher activity, given that higher NOS activity observed in humans is due to different genes (vs. splice variants of the same gene) and that NOS activity in humans is of a different order of magnitude than implied here for beewolves. The inhibition experiment provides some evidence for NOS being involved, although there is still the assumption that NOS is the only enzyme affected by the inhibitor. Differential gene expression matching NO production, direct assays of the NOS variants, or even signatures of adaptive evolution (e.g., Dn/Ds metrics) would provide more convincing support for the uniqueness of the beewolf NOS, especially when combined with explicit comparisons to related species that due not produce similarly high levels of NO as controls.

We agree that there might have been too much focus on the NOS gene and the alternative splicing as an explanation of the elevated NO production rates. We have now changed the text to be more cautious, stated that there might be another, yet unknown pathway and conclude with the hypothesis that the NOS is an important component of NO production in beewolf eggs. However, we have now obtained a draft genome sequence of *P. triangulum*, and there is only one Pt-NOS gene, confirming our finding from transcriptomic analyses. The genome is currently subject to careful annotation and evolutionary analyses. In addition, gene expression analyses are included in the manuscript (Figure 10, Figure 7 in the revised manuscript), and the peak in NOS transcripts corresponds to the peak in NO production at the respective temperature, providing strong support for the production of NO from the single NOS that occurs in beewolves.

Reviewer #3:This is a very nice paper that touches on different fascinating aspects of insect's biology, from an ecological to a molecular point of view, and should interest a broad audience.This study shows that eggs of the parasitic wasp Philanthus triangulum emit high amounts of nitric oxide as antifungal compounds to protect themselves and preserve the food source of their progeny. The enzyme responsible for NO production is identified (NOS) and a differential splicing event between egg and adult forms may explain the high levels observed in eggs. These are novel and quite interesting findings that complement other defense strategies that have been previously reported in insects. The story is well written and nicely presented. Data are convincing. The Discussion is complete and carefully addresses the novelty and the biochemical and biological implications of the findings.That eggs emit a volatile compound that has antifungal activity is demonstrated by 1) showing that host bees not in direct or indirect contact with P. triangulum eggs get more infected with mold fungi, 2) showing that eggs release an antifungal volatile when placed above a culture of different fungal strains growing on agar plates. That eggs emit NO is demonstrated by histochemical and colorimetric methods and by using fluorescent probes. However, that egg-released NO is responsible for inhibiting fungal growth is only supported by the correlation between the above observations and by previous reports on the mycotic activity of NO. I see the difficulty to genetically prove the causality of NO production in the antifungal effect of egg-derived volatile production. A simple additional experiment may however help to strengthen this hypothesis. It would be important to inhibit NO release in eggs by using L-NAME, the NOS-inhibiting substance used for Figure 6, and to show that egg-induced inhibition of fungal growth on agar plate (as in Figure 3) is abolished or greatly diminished.

As stated above, we have conducted an alternative additional experiment that shows that synthetic NO in concentrations similar to those estimated for brood cells exert a similar effect on fungus growth as the gas emitted by eggs.

[Editors' note: further revisions were requested prior to acceptance, as described below.]In response to the reviews you have carried out an additional experiment and have modified the manuscript in various places. As a result the manuscript has been clearly improved. Yet, a few issues still remain.The additional experiment tests the effect of synthetic NO on fungal infection of honeybees and shows that synthetic NO protects the bees from infection. This experiment was done for logistic reasons as an alternative to the suggested experiment (eliminating NO· from the natural mixture with the inhibitor). The result underlines the involvement of NO but leaves the option of other compounds being involved open. One of the reviewers had commented that the focus on NO is quite directed, and so might result in missing other potential volatile compounds that are also involved in the observed bioactivities. In response to the question to better justify the focus on NO you comment that is not reasonable. If the suggested experiment had been done that might be the case but because the involvement of other compounds has not been excluded I consider it reasonable to explain why you focussed on NO.

We have now added information about the considerations that led us to focus on NO/NO_2_. Besides the typical odor of oxidizing gases, we tested the occurrence of an oxidant using iodine/starch solution. The positive reaction clearly showed that there was a strong oxidant. Most of the possible candidates were rather unlikely since they were –to our knowledge- not known to be produced by organisms (O_3_) or their occurrence was restricted to special processes (Cl_2_). Since NO was known as a ubiquitous biological effector and its release would generate the typical odor (due to its oxidation to NO_2_), we focused on NO and conducted the relevant analyses, including the specific Griess assay. Our additional experiment shows that concentrations of synthetic NO similar to natural brood cells have an antimycotic effect that does not deviate from that of the gas emitted by the beewolf eggs. Nevertheless, we cannot exclude that the egg releases small amounts of other volatiles. This is stated now.

In response to the comment that "the observation that larvae are not resistant to NO produced by the eggs is concerning." you respond with comments in the response letter but this has not resulted in a change in the manuscript. It will be important to provide information in the manuscript itself why the larvae will not suffer from NO as this is vital to the understanding of the protective activity of the NO in conserving a resource. This aspect was also identified by one of the other reviewers.

Actually, we had made some changes, however we have now rephrased the text to more explicitly explain, why larvae do not suffer from the high concentrations of NO that are generated by the eggs.